# Regional opportunities for tundra conservation in the next 1000 years

Stefan Kruse[1]*, Ulrike Herzschuh[1,2,3]

[1]Alfred Wegener Institute Helmholtz Centre for Polar and Marine Research, Potsdam, Germany; [2]Institute of Environmental Sciences 6 and Geography, University of Potsdam, Potsdam-Golm, Germany; [3]Institute of 7 Biochemistry and Biology, University of Potsdam, Potsdam-Golm, Germany

**Abstract** The biodiversity of tundra areas in northern high latitudes is threatened by invasion of forests under global warming. However, poorly understood nonlinear responses of the treeline ecotone mean the timing and extent of tundra losses are unclear, but policymakers need such information to optimize conservation efforts. Our individual-based model LAVESI, developed for the Siberian tundra-taiga ecotone, can help improve our understanding. Consequently, we simulated treeline migration trajectories until the end of the millennium, causing a loss of tundra area when advancing north. Our simulations reveal that the treeline follows climate warming with a severe, century-long time lag, which is overcompensated by infilling of stands in the long run even when temperatures cool again. Our simulations reveal that only under ambitious mitigation strategies (relative concentration pathway 2.6) will ~30% of original tundra areas remain in the north but separated into two disjunct refugia.

## Editor's evaluation

This study will be of interest to researchers studying climate change and Arctic tundra systems. The authors apply LAVESI, a machine-intensive and spatially explicit simulation of individual Siberian trees at the tundra-forest boundary, to call attention to the rapid reduction in the tundra biome as climate warming pushes forests toward the Arctic Ocean. This detailed modelling study predicts dramatic losses of tundra area by the middle of the millennium even under an ambitious climate mitigation scenario and highlights considerable risks of extinction.

*For correspondence:
stefan.kruse@awi.de

Competing interest: The authors declare that no competing interests exist.

## Introduction

Arctic regions have experienced strong warming over recent decades (*Box et al., 2019*). The warming is expected to continue drastically based on future changes simulated with prescribed 'relative concentration pathway' (RCP) scenarios (*Meinshausen et al., 2011*). Following climate simulations, mean July temperatures in the region could be, by the end of the 21st century, less than +2°C warmer with ambitious mitigation efforts (RCP 2.6), about +3.1°C (intermediate effort scenario RCP 4.5), or an extreme of +14°C warmer (high-emission high-warming RCP 8.5). In consequence, the treeline ecotone can expand and will squeeze Arctic regions from the south and have only limited refugial potential in the north (*Hofgaard et al., 2012*; *MacDonald et al., 2007*) as simulation studies predicted (*Rupp et al., 2001*). This turnover threatens the existence of various types of sensitive tundra with differing vegetation structure along latitudinal and elevational bioclimatic gradients located between the treeline in the south and the Arctic Ocean in the north (*Walker et al., 2005*; *Pearson et al., 2013*; *Yu et al., 2009*). The Siberian tundra areas are known for their high regional floristic diversity and species biodiversity (*Pauli et al., 2012*; *Mod and Luoto, 2016*; *Arctic Climate Impact Assessment,*

2004; Schmidt et al., 2017) and support special (indigenous) types of land use (e.g. reindeer herding, foraging). However, the expansion of forests in response to increasing temperature is not well understood and the uncertain timing of future tundra loss is challenging conservation efforts.

In Siberia, the climate velocity of isotherms within the tundra biome has been calculated as $\sim 290\ \mathrm{m\ year^{-1}}$ (Loarie et al., 2009). When assuming a direct equilibrial relationship linking the modern position of the treeline to its main limiting factor of July temperature (MacDonald et al.,

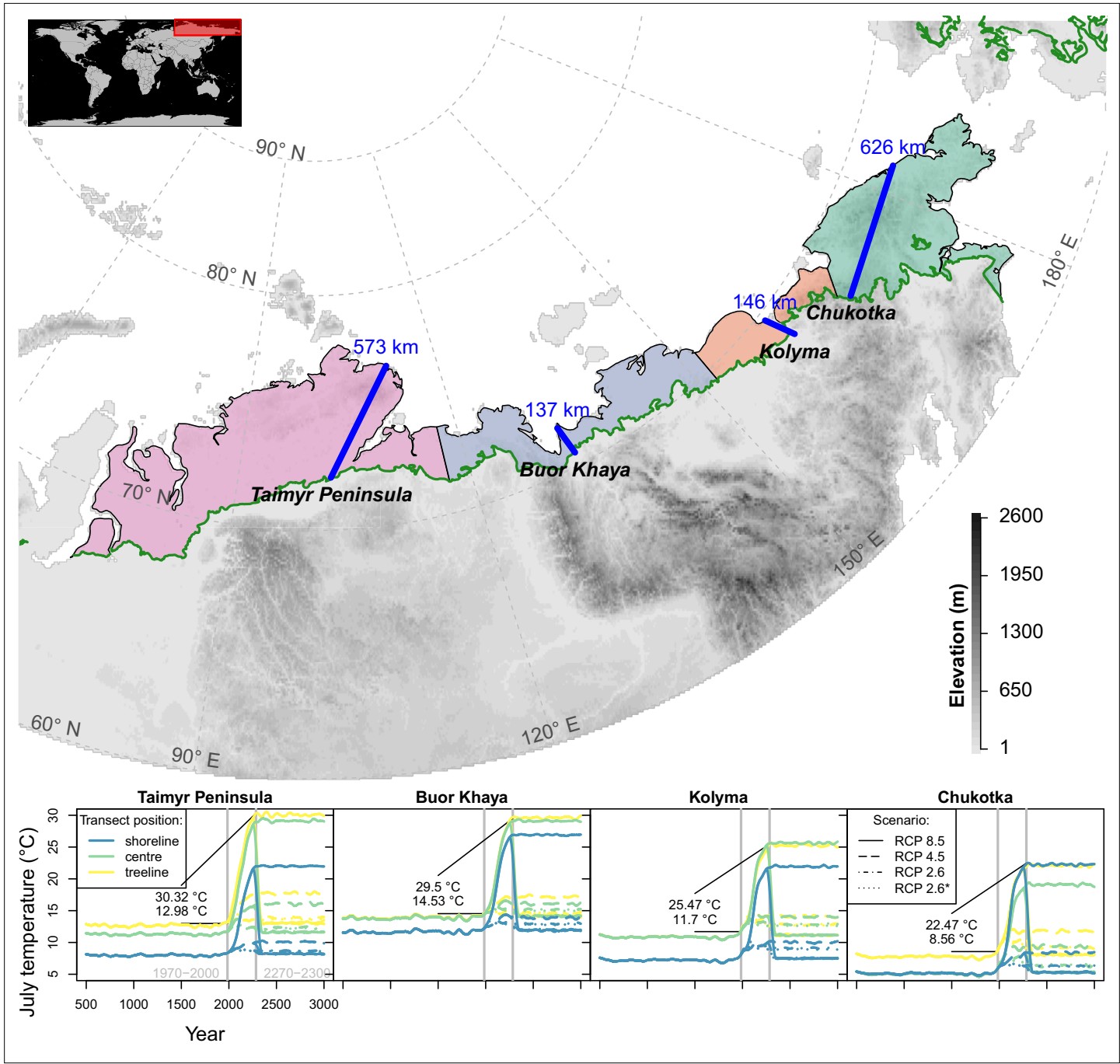

**Figure 1.** Transects (blue lines) were placed starting in the treeline at field sites and extending to the shoreline of the Arctic Ocean (map); only tundra areas above the treeline (Walker et al., 2005) are considered. The land area was grouped into four regions equidistantly separated between the transects (lower plot) and plots show climate forcing for 501–3000 CE based on relative concentration pathway (RCP) scenarios in these regions at the extremes and middle part of the transects; numbers give mean values for modern and future strongest warming under RCP 8.5. Map projection: Albers Equal Area.

*2007*), the treeline is expected to reach its maximum expansion within the first centuries of the millennium. Regionally, where the shoreline of the Arctic Ocean is close to the treeline, the 100–200-km-wide tundra corridor would be completely covered by forests (*Figure 1*). Moreover, even regions with wide tundra corridors in Siberia (~ 600 km; Taimyr Peninsula and Chukotka) could be forested in the warmest scenarios. However, the potential migration rates are likely constrained by other factors, partly acting on a small scale (*Corlett and Westcott, 2013*). This and the fact that in Siberia the treeline is formed solely by larch species growing on permafrost while in North America this niche is covered by a mix of species (*Mamet et al., 2019*; *Herzschuh and Jordan, 2019*) make it difficult if not impossible to transfer knowledge from any other treeline region to inform Siberian tundra responses.

Equilibrium-based estimations are not backed by empirical studies which are showing slower responses and either no treeline advance or rates of only a few metres per year in Siberia (*Scherrer et al., 2020*; *Kharuk et al., 2018*; *Wieczorek et al., 2017b*; *Rees et al., 2020*). Either the different response of regions can be related to site conditions (e.g. *Rees et al., 2020*; *Mamet et al., 2019*) or it is not well understood how complex processes such as seed dispersal or intra-species competition acting on fine and broad scales constrain the response to warming (*Corlett and Westcott, 2013*; *Wieczorek et al., 2017b*). Models that do not consider these processes lead to forecasts prone to overestimating treeline advance (compare *Pearson et al., 2013*; *Snell and Cowling, 2015*), but it is challenging to consider both small-scale processes and the broad-scale picture.

Models which contain processes to estimate competition between individuals for resources and seed dispersal in a spatially explicit environment and which handle each tree individually are called individual-based spatially explicit models (*DeAngelis and Mooij, 2005*). These models have become increasingly used lately to assist in formulating conservation management strategies (*Zurell et al., 2021*). For the northern Siberian treeline ecotone, the vegetation model LAVESI was developed, which incorporates a full tree (*Larix*) life cycle beginning with seed production and dispersal, germination, and establishment of seedlings, which then grow based on abiotic (temperature, precipitation, active-layer depth) and biotic (competition) conditions (*Kruse et al., 2019a*, *Kruse et al., 2016*). Although soil nutrients and mycorrhiza are not considered explicitly in the current version, it was shown that it can capture the nonlinear response of the treeline populations well at a regional level on the Taimyr Peninsula (*Kruse et al., 2019a*, *Kruse et al., 2016*; *Wieczorek et al., 2017b*).

To cope with the complex forest-tundra ecotone responses, which are not in equilibrium with climate (*Hofgaard et al., 2012*), we updated and made use of our individual-based spatially explicit model LAVESI. With our simulation study, we explore treeline migration and tundra area dynamics to provide important information as to which tundra areas can survive future warming and where preventive measures for protection would be necessary to conserve the unique Siberian tundra. This study is guided by two questions: 'What are the treeline trajectories in Siberia for different climate mitigation

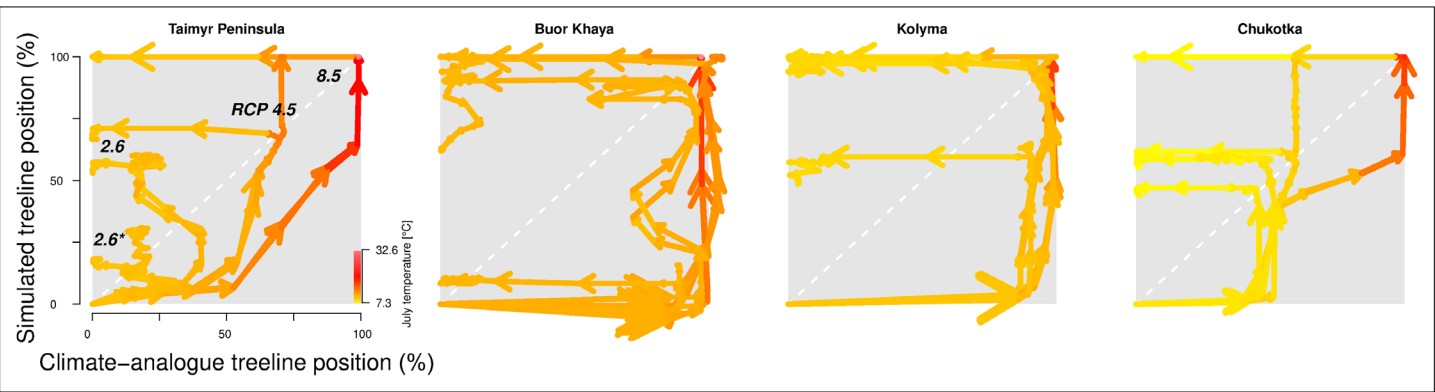

**Figure 2.** The trajectory of the simulated treeline position relative to its current position and the maximum at the shoreline versus its climate-analogue position for the four regions shows a migration lag of the treeline during the first centuries (each line segment represents 25 years and the length of the arrow head corresponds to the step length) until the simulated treeline is limited by climate. Forests expand their area further and infilling proceeds when climate conditions cool and even overshoot in the long run with cooling back to 20th-century temperatures. The diagonal is where climate and the treeline are in equilibrium; below the diagonal, tree migration lags climate; above the diagonal is 'overshooting' and reaching locations actual climate would allow. For each relative concentration pathway (RCP) scenario, two are presented, one for the scenario as-is and the second for the cooling; scenario RCP 2.6* warms only at half the rate of RCP 2.6. See also *Appendix 3—figure 1* for the year when the trajectory passes the equilibrium.

scenarios (RCP)?' and 'Under which climate scenario can tundra retreat to refugia and recolonize areas after temperatures cool again, and where?'.

## Results and discussion

### Treeline migration trajectories under future climate changes

The simulations revealed that the treeline is not in an equilibrial relationship but its advance follows climate warming, potentially reaching $30\,\mathrm{km\,decade}^{-1}$ in the 21st century under warmest climate scenarios (*Appendix 2—table 1*). When relating and tracking the position of the treeline to the climate (=climate-analogue), we find a severe migration lag of hundreds of years until simulations reach the same position (*Figure 2*). This behaviour is longer than expected from the properties of treeline-forming larch species with generation times of decades and weak long-distance dispersal rates (e.g. *Mamet et al., 2019*; *Kruse et al., 2019a*; *Scherrer et al., 2020*; *Clark, 1998*). Other processes, not yet explicitly captured by the model, may constrain the migration process, such as the strong greening of the Arctic increasing interspecific competition (compare *Pearson et al., 2013*; *Berner et al., 2020*), the warming and abrupt thaw of permafrost (e.g. *Stuenzi et al., 2021*; *Biskaborn et al., 2019*), soil nutrients (e.g. *Sullivan et al., 2015*), and animal activity (e.g. *Wielgolaski et al., 2017*). Regardless, our simulated rates over the past few decades are generally in line with the few existing modern observations showing only a slow treeline advance (*Mamet et al., 2019*; *Hansson et al., 2021*; *Rees et al., 2020*; *Shevtsova et al., 2020*; *Guo and Rees, 2020*). Rates are especially high in Chukotka, however, in all forcing scenarios (*Appendix 2—table 1*): an observation that can be attributed to forest patches ahead of the current treeline, which serve as nuclei for tundra infilling (shown by, e.g. *Snell and Cowling, 2015*; *Kharuk et al., 2018*; *Kruse et al., 2019a*; *Harsch and Bader, 2011*). Generally, the simulated rates are high compared to those during post-glacial migration (e.g. *Feurdean et al., 2013*) but seem likely as future climate is unprecedented, especially in their rate of change.

After the initial migration lag, we can see from our millennium-long simulations that the treeline reaches more advanced positions than the climate analogue (*Figure 2*, *Appendix 2—table 1*, *Appendix 2—figure 1*, *Appendix 2—figure 2*, *Appendix 3—figure 1*). This strong overshooting was unexpected especially in scenarios with cooling temperatures back to the 20th-century level. However, we designed our model to be pattern-oriented and is based on observations that, once

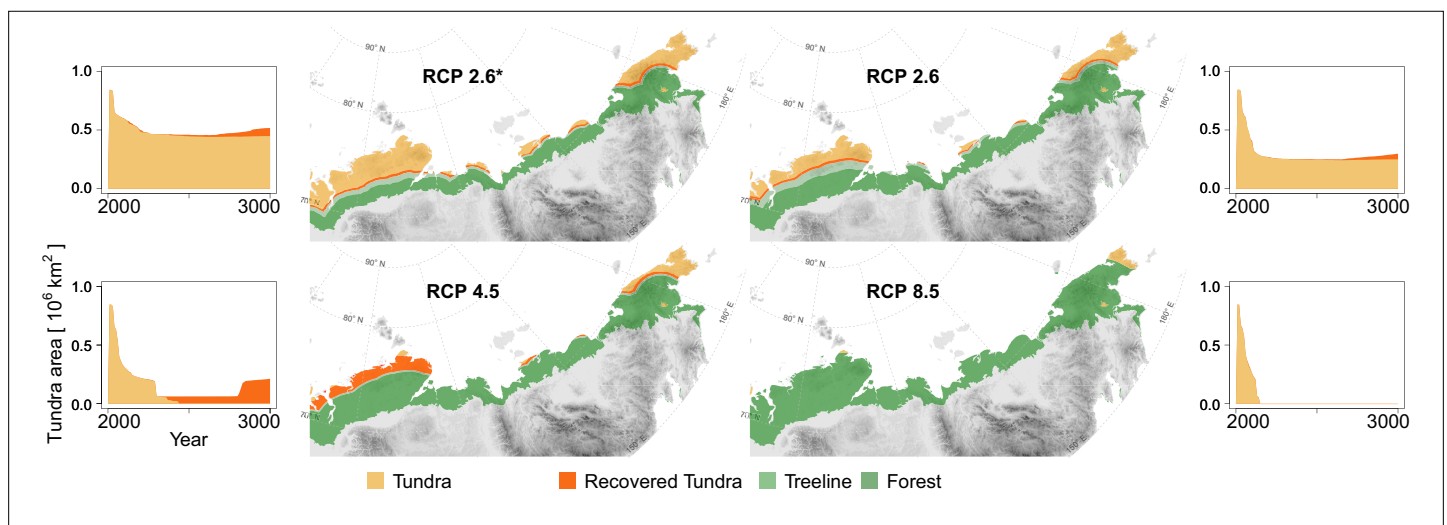

**Figure 3.** Forest position and tundra area at year 3000 CE for different climate mitigation scenarios and under potential cooling back to 20th-century temperatures after peak temperatures have been reached. The area of tundra changes over time and can only partly recover when temperatures cool and forests recede (plots next of the maps show years 2000–3000 CE). Only tundra areas above the treeline (*Walker et al., 2005*) are considered. Map projection: Albers Equal Area.

The online version of this article includes the following video for figure 3:

**Figure 3—video 1.** The development of the forest position and tundra area can be seen in a video supplement presenting the state in 10-year steps.
https://elifesciences.org/articles/75163/figures#fig3video1

established, trees can endure periods of unfavourable environments (compare *Kruse et al., 2016*). This leads to the survival of trees ahead of the treeline acting as nuclei for tundra colonization when conditions improve (*Väliranta et al., 2011*; *Snell and Cowling, 2015*). This is backed by observations of slower-than-expected retreat of elevational treelines because trees can persist until dramatic events cause the death of all well-established trees (*Scherrer et al., 2020*).

## Tundra area dynamics caused by forest expansion

The area loss over time roughly follows a negative exponential function, which is steeper the warmer the climate scenarios are (*Figure 3*, *Appendix 3—figure 1*). The quickest loss, already by ~2100 CE, is seen in the intermediate warming scenario RCP 4.5 (+3°C July temperatures) and the extreme RCP 8.5 (July temperatures +14°C). For these scenarios, tundra retreats to its minimum cover of 5.7% by the middle of the millennium (lower panels in *Figure 3*). Even under mitigation scenario RCP 2.6, the tundra area is reduced to 32.7% or in the very ambitious scenario 2.6* to only 54.9%. Other simulation studies show 50% loss during the 21st century (*Callaghan et al., 2005*; *Kaplan et al., 2003*; *Pearson et al., 2013*) but not the dynamics of the long run until the end of 3000 CE, as shown by our study.

Cooling back to the 20th century levels led to a slight tundra re-expansion of a few percent of the area by the end of the simulations. While re-expansion is small for RCP 2.6* and 2.6, and largest for RCP 4.5 on the Taimyr Peninsula where forests retreat from areas of $\sim 200,000\,\text{km}^2$ in the final 200 years of the simulation, under RCP 8.5 the effect of the cooling led to no tundra releases. In consequence, only if humans act ambitiously in limiting global warming to well below +2°C (RCP 2.6, 2.6*), and temperatures actually cool after the maximum, do simulations show that ~10–30% of the tundra area could persist in the north and can re-expand after the mid-millennium maximum treeline extent. However, due to the overshooting effect, less than 10% of the area remains unforested. All in all, the expansion is in accordance with other studies and will likely put especially cold-climate tundra types at risk of extinction (e.g. *Pearson et al., 2013*; *Walker et al., 2005*).

## Concluding opportunities for tundra conservation

Following our simulation results, tundra is reduced to ~5.7% of its current area at maximum forest expansion under the most likely scenario RCP 4.5 after an initial migration lag. Only ambitious strategies to mitigate climate warming (e.g. RCP 2.6) could prevent such an extensive loss and retain ~32.7% of the current tundra. In both cases, the northern refugia are located in Chukotka and the Taimyr Peninsula, currently covered by High Arctic and Arctic tundra types which have relatively high biodiversity (*Arctic Climate Impact Assessment, 2004*). A WWF report (*Krever et al., 2009*) finds that the current total cover of protected northern areas is insufficient to achieve the requisite 30% protection necessary for biodiversity conservation (*Secretariat of the United Nations Convention on Biological Diversity, 2021*). Especially cold-climate tundra types on Siberian islands and in Chukotka are underprotected (e.g. *Pearson et al., 2013*; *Walker et al., 2005*) and thus further protected areas need to be established.

Although the large refugia could enable tundra species survival and later recolonization (analogon forests, *Harsch et al., 2009*; *Clark, 1998*; *Stralberg et al., 2020*), the diversity in these ~2500 km distant fragments is threatened by the fundamental disadvantages of a small population (inbreeding/ genetic drift, *Ohsawa and Ide, 2008*; *Charlesworth and Charlesworth, 1987*; *Mona et al., 2014*; *May et al., 2013*). A network of connected systems would be necessary with protection of potential refugia (e.g. *Morelli et al., 2020*) distributed across the ~4000-km-wide tundra area. Due to its individual-based nature, LAVESI can be a tool to assess population genetics and aid in the optimal placement of migration corridors and areas to protect (cf. *Zurell et al., 2021*).

# Materials and methods
## Model description

The model LAVESI is an individual-based and spatially explicit model that was developed to study the treeline ecotone growing on permafrost soils on the Taimyr Peninsula (*Kruse et al., 2016*; *Wieczorek et al., 2017b*; *Epp et al., 2018*) and since then it has been improved and modified for different purposes including for migration rate studies (*Kruse et al., 2019a*). To achieve the most realistic model, each life history stage of the larch individuals is handled explicitly, and the model's processes

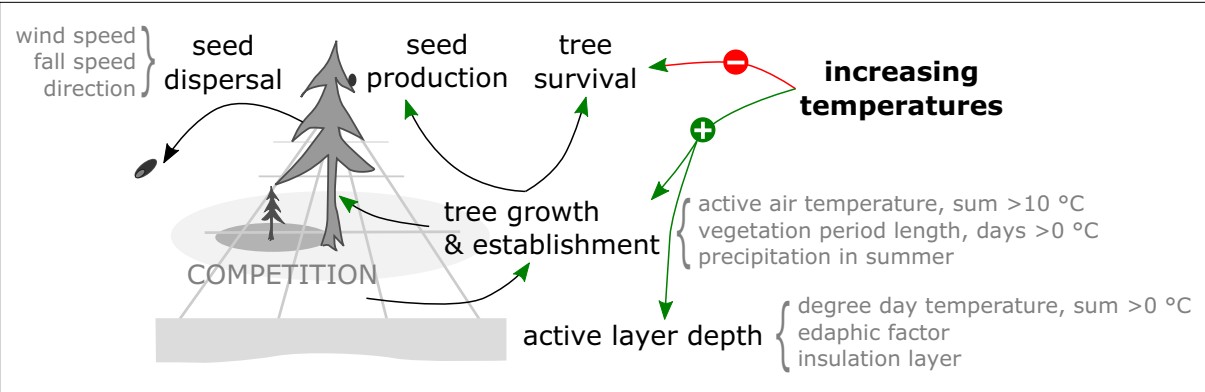

**Figure 4.** The model LAVESI invokes in each yearly time step processes to individually handle seeds and represent the full life cycle of simulated trees, leading to forest stand dynamics. The climate environment drives establishment, growth, and mortality.

adapted to observed patterns of surveyed tree stands for the dominant larch tree species, *Larix gmelinii* and *Larix sibirica*.

The simulation proceeds in yearly time steps from the beginning to the end of the input climate time series following a stabilization period to ensure that emerging populations reach equilibrium with the environment. Stochasticity in the model was introduced by using random numbers generated with a pseudo random number generator.

Within one simulation year, the following processes become consecutively invoked (*Figure 4*):

*Update of environment*: Interactions between neighbouring trees are local and indirect. Basal diameters of each individual tree are used to evaluate the competition strength. We use a yearly updated density map to pass information about competition for resources between trees. An active-layer depth is estimated based on an edaphic factor and the number of days exceeding 0°C following simplifications by *Hinkel and Nicholas, 1995*.

*Growth*: The maximum possible growth is estimated based on 10 years mean climate auxiliary variables (temperature of the coldest [January] and warmest [July] month, $NDD_0$ 'net degree days', $AAT_10$ 'active air temperature'). The individual growth of basal diameter and, if a tree reached a height of 1.3 m, breast height diameter is calculated from the maximum possible growth in the current year affected by the tree's density index. From the resulting diameters, the tree height is estimated.

*Seed dispersal*: Seeds in 'cones' are dispersed from the parent trees. The dispersal directions and distances are randomly determined from a ballistic flight influenced by wind speed and direction with decreasing probabilities for long distances, and, if dispersed seeds leave the extent of the simulated plot, they can be introduced from the other side only on the east–west margins.

*Seed production*: Trees produce seeds after the year at which they reached their stochastically estimated maturation height. The total amount depends on weather, competition, and tree size.

*Establishment*: The seeds that lie on the ground germinate at a rate depending on current weather conditions.

*Mortality*: Individual trees or seeds die, that is, they become removed from the plot, at a specified mortality rate. For trees this is deduced from long-term mean weather values, a drought index, surrounding tree density, tree age and size, plus a background mortality rate. Seeds, on the other hand, have the same constant mortality rate whether on trees or the ground.

*Ageing*: Finally, the age of seeds and trees increases once a year and seeds are removed from the system when they reach a defined species age limit.

Many factors influence the success of tree establishment and growth, of which the following seem most important to understand treeline migration. Warmth, especially summer mean temperatures, cumulative temperatures, and the vegetation period length, are key climate variables (e.g. *MacDonald et al., 2007*) and are hence included in the model for estimating potential tree growth and tree mortality (*Kruse et al., 2016*). This is further influenced by the abiotic (permafrost, active-layer depth) and biotic (intraspecific competition) environment. For example, permafrost soils with a shallow active-layer depth limit the space available for rooting and nutrient stores (e.g. *Sullivan et al.,*

*2015*) and frozen/cold soils prevent growth but water can support growth in dry summers (*Sugimoto et al., 2002*). The latter functioning could be lost in future hot environments, and in consequence it would decrease the treeline advance rate further and thereby increase the lag time. Accordingly, the active-layer depth is explicitly simulated in the model's environment module, but soil nutrients are considered only implicitly. Individuals compete for space which serves as a surrogate for this. Permafrost degradation/abrupt thaw and warming (*Biskaborn et al., 2019*; *Stuenzi et al., 2021*) could lead to thermokarst processes that would locally cause swamping/waterlogging, leading to suppressed establishment/growth (*Rees et al., 2020*), which is not explicitly simulated and unlikely to limit tree invasion on a broader scale. Further limitations could arise from the lack of a suitable mycorrhiza symbiosis partner, but as shown by *Hippel et al., 2021* even in northernmost areas on the Taimyr Peninsula mycorrhiza are present even before the invasion of trees in the Holocene. Additionally, dispersal rates of mycorrhiza with small spores are a magnitude faster than for taxa with larger seeds so that geographic distance is unlikely to be a limiting factor, similar to diatoms as shown by *Stoof-Leichsenring et al., 2015*. On the other hand, interspecific competition of recruits with present shrubs or grasses which are increasing more quickly – called the greening of the Arctic (*Pearson et al., 2013*; *Berner et al., 2020*) – is not explicitly simulated in LAVESI, although it is partly included in the germination probability and survival rate of individuals at the seedling stage. Further biotic interactions are implicitly included, such as herbivory by browsers (e.g. *Wielgolaski et al., 2017*), which reduces the survival and growth rate by damaging the shoots (compare krummholz study by *Kruse et al., 2020*) and can lead to a snow compaction cooling effect on soil temperatures (*Beer et al., 2020*), or pest outbreaks that are likely to increase in the future (e.g. larch silk moth, *Fedotova et al., 2019*). This would most likely lead to a further reduction of migration rates.

## Model improvements

The source code was updated to LAVESI-WIND version 1.2 on the crutransects branch to be sufficiently light in terms of memory allocation (details in Appendix 1) and including landscape (developed in parallel for Chukotka mixed alpine-latitudinal treeline simulations in *Shevtsova et al., 2021*; *Kruse et al., 2021*). The model is freely available on GitHub (https://github.com/StefanKruse/LAVESI; *Kruse et al., 2022*). The source code of the version used in this article is permanently stored on Zenodo (https://doi.org/10.5281/zenodo.6344261).

## Transect setup

At four representative locations of the Siberian treeline within 100–170° E, we placed linear transects, starting at a field site, and allowed open tree stands to establish up to the shoreline (*Figure 1*). From west to east: (1) Taimyr Peninsula, 573 km long, site 13-TY-09-VI, 72.15067° N (*Wieczorek et al., 2017b*, *Wieczorek et al., 2017c*), (2) Buor Khaya Peninsula, 137 km long, 14-OM-20-V4: 70.52671° N (*Liu et al., 2020*), (3) Kolyma River Basin, 146 km long, 12-KO-02/I: 68.38916° N (*Wieczorek et al., 2017a*), and (4) Chukotka, 626 km long, EN18022: 67.40102° N (*Kruse and Stoof-Leichsenring, 2016*; *Kruse et al., 2019b*). While the first three locations in the west cover predominantly flat terrain, the easternmost location in Chukotka has mountainous stretches.

## Climate forcing

Temperature and precipitation time series were constructed using the following steps:

1. Extract climate data in 10 km steps along transects beginning at the position of the field sites for each of the four focus regions from the grid cells of CRU TS 4.03 by distance-weighted interpolation (*Harris et al., 2020*).
2. Establish a yearly $\delta^{18}$O and ice layer thickness series for 501–1900 from the yearly resolution ice-core data from Severnaya Zemlya (north of the Taimyr Peninsula, available for years 934–1998; *Opel et al., 2013*), the only closest annual resolution archive allowing both temperature and precipitation reconstructions independent of plant-based proxies (pollen or tree rings). This was achieved by sampling 25-year-long blocks of yearly resolution data from the pre-industrial (< 1800 CE) and adding this variability to the AN86/87 core which dates back to 4703 BCE (*Arkhipov et al., 2008*).
3. Adjust this locally to the 10 km steps climate data series per transect by building linear models of yearly temperature and precipitation for the overlapping years 1901–1998 and use this to estimate mean values for the period 501–1900.

4. Sample 15-year-long blocks from CRU and adjust the mean temperature by the difference of the sampled block from the estimated mean values to achieve monthly data series.
5. Prolong this series by coupled climate model long runs until 2300 CE with model output with MPI-ESM-LR prepared for CMIP5 for scenarios RCP 2.6, RCP 4.5, and RCP 8.5 and additionally a scenario that warms only at half the rate of RCP 2.6, which is here called RCP 2.6*.
6. Extend these trends for 2100–2300 until 2500 CE and repeat in the following either the climate data of 1901–1978 in a loop, which in this study is called 20th-century cooling, or copy the period 2300–2500 until 3000 CE.

July temperatures (*Figure 1*), which have the strongest impact in the model due to a positive influence on tree diameter growth, increase in these scenarios by a median of +1.2°C (RCP 2.6*), +1.8°C (2.6), +2.1°C (4.5), and up to +5°C (8.5) compared to 1970–2000 until 2100, which partly cools until 2300 to +0.6°C (RCP 2.6*), +0.5°C (2.6), and reaches higher levels for the two warmest scenarios to +3.1°C (4.5) and up to +144°C (8.5).

Wind speed and direction data were obtained at 6-hourly resolution for 1979–2018 from EraInterim5 (dataset description in *Dee et al., 2011*) for each of the transects and supplied with the model for estimation of seed dispersal events. In a prior version, these data were only available for the Taimyr Peninsula and only for years until 2012.

Future climate changes will lead to an increase in extreme weather events (*Masson-Delmotte et al., 2021*), among others a higher ignition probability and thus wildfires are expected to which the Siberian boreal forests are sensitive (e.g. *Shvidenko and Schepaschenko, 2014*). Fire is not yet explicitly simulated in the model and in a hotter environment fires could lead to dramatic forest losses. Areas might be turned into steppe and not revert to the prior tundra. Increases in extreme rainfall events will lead to local flooding and waterlogging (*Masson-Delmotte et al., 2021*), a response not explicitly implemented in the model. The anticipated effect to the LAVESI predictions is that tundra colonization would most likely be slowed down in the lowlands but less so in the well-drained mountainous regions, which would further prolong the time-lagged response of the treeline migration.

## Tree growth on transects

We implemented sensing of the environment for each individual tree via its y-position along the transect which is made up of 10 km spaced climate data. The maximum possible growth at a certain position (see initial publication *Kruse et al., 2016*) is calculated by interpolating the possible growth under the climate using the two closest climate variables.

The larch species *L. gmelinii* (Rupr.) Rupr. dominates the areas to the west of the Verkhoyansk Mountain Range ~90–130° E and its sister species *L. cajanderi* Mill. grows in northeast Siberia *Figure 1* (*Abaimov, 2010*). Hence, the already implemented species *L. gmelinii* was set to be present in the simulations and simulations tuned to fit observations. To initiate growth 100,000 seeds were introduced at the cold end with a distance based on a negative exponential kernel for the first 50 years. Similarly, extinction on a transect was prevented by permanently introducing 1000 seeds per 200 km length. Further seed introduction from the nonexplicitly simulated hinterland was estimated on a 500-m-long stretch at the cold site linking the production and release height to climate.

The model used for this study is solely forced by climate data and no vegetation feedback on permafrost soils and climate was implemented. Therefore, the expected decrease of the albedo leading to a positive feedback with climate when tundra is colonized by tree stands (e.g. *Pearson et al., 2013*) is neglected here. Considering this would likely further increase the rate of transition from tundra to forests.

## Model tuning and validation

Simulation runs were forced with the prepared climate data for the transects for validation of the shape of the treeline as simulated by the model. The results of the year 2000 were compared to field data that was positioned along the temperature gradient on the simulated transects by using the respective CRU TS temperature data for each grid cell of each plot (example *Figure 1*). The field data collection included disturbed sites (e.g. in Chukotka), leading to very high stem counts, which are the number of trees exceeding 1.3 m in height per hectare. The model could be tuned by introducing a local adjustment based on elevation lapse rates of the relevant climate values used internally, which is necessary as the climate data may lead to over-/underestimations locally in a model that does not

consider local topography: an issue that is especially relevant in the mountainous region of Chukotka. We made a moderate fit with the following amendments:

- Taimyr Peninsula: $T_{Jan} = -1.17°C$, $T_{Jul} = -0.84°C$ and $P_{Year} = -5.4$ mm
- Buor Khaya Peninsula: $T_{Jan} = -0.23°C$, $T_{Jul} = -2.48°C$, and $P_{Year} = -28.9$ mm
- Kolyma River Basin: $T_{Jan} = +3.14°C$, $T_{Jul} = +1.26°C$, and $P_{Year} = +85.7$ mm
- Chukotka: $T_{Jan} = +4.46°C$, $T_{Jul} = +4.30°C$, and $P_{Year} = +8.2$ mm

From the individuals present along the simulated transects, the northernmost positions of the treeline are calculated. For the four focus regions these are +66, +88, +98, and +16 km beyond the treeline start location at the year 2000 (*Appendix 2—figure 2*), which are partly ahead of the observed positions based on visual inspection of satellite data ~+30, +30, +80, and +15 km beyond the field location selected as the starting point.

We assume that the fundamental drivers of tree growth remain stable in time. The space-for-time approach used here is based on using several sites from across the treeline in the regions of interest for model parameterization and validation that have experienced a range of past climate (*Likens, 1989*). However, climate warming will lead to unprecedented temperatures (*Figure 1*) that will affect the current stand structure and distribution. This may cause changes in the response of single considered processes or make other processes that are not currently important necessary (e.g. *Blois et al., 2013*; *Likens, 1989*). Keeping this in mind, the future simulations should be interpreted with caution, although a simulation study with LAVESI revealed realistically predicted migration rates and patterns for the recent decades of strong warming in the study region of Chukotka (currently in review).

## Simulation setup and data processing

We ran three simulation repeats for each transect with the climate series prepared for this study as initial tests showed robust simulation results. To consider the available area from the start to the shoreline, we used 800-km-long and 300-km-long transects and 20-m-wide wrapping for the east and west boundary of each simulated stretch, thereby representing a slice of a continuous treeline.

The positions of three key variables of stand densities expressed in stems, which are trees >1.3 m tall, are extracted in 10-year steps: single-tree stands, defined as the northernmost position of forest islands ahead of the treeline with $\geq 1$ stem ha$^{-1}$; treeline stands, the northernmost position of a continuous forest cover with $1$ stem ha$^{-1} \leq$ density $< 100$ stem ha$^{-1}$; and forestline stands, the northernmost position of forest cover $> 100$ stems ha$^{-1}$ (see Figure 2 in *Kruse et al., 2019a* for a graphical representation). The position of the simulated treeline determined for the year 2000 is used as the baseline for the calculation of migration rates. The resulting values for each of the three key variables are used in 10-year steps to interpolate with a weighted average the expansion starting at the current treeline (e.g. *Walker et al., 2005*). This is similar to the biotic velocity of *Ordonez and Williams, 2013*. Opposed to this, we extrapolated the position of the modern observed treeline based on July temperatures (=climate-analogue) for the years 2000–3000 CE. Therefore, the current July temperature at the treeline position is determined and tracked along the transect at each time step of the climate forcing, which is similar to the climatic velocity of *Loarie et al., 2009*.

We used R version 3.6.1 (*R Development Core Team, 2019*) for data handling, statistical analyses, and graphical representation. The 30 arcsec WorldClim 1.4 data (*Hijmans et al., 2005*) along the transects and 2.5 min of a degree elevation for mapping was downloaded through the geospatial package raster version 3.0-12. (*Hijmans, 2020*). Additional geospatial packages sf version 1.4-2 and rgdal version 1.4-7 (*Pebesma, 2018*; *Bivand et al., 2019*) were used to reproject the treeline shape to Albers Equal Area projection and calculate weighted average buffers at each 10-km step along the treeline and merge the buffer polygons for each of the advancement values for the three key variables (single trees, treeline, forestline). The treeline constructed by the circumarctic vegetation map consortium (CAVM, *Walker et al., 2005*) is provided in segments that were simplified in QGIS version 3.10 (*QGIS Development Team, 2020*). For tundra area assessment, the land area polygons of Russia accessed via R package maps version 3.3.0 (code by *Richard and Becker, 2018*) were cleaned and the islands in the Arctic Ocean excluded. Linear models for temperature and precipitation reconstruction and climate-lapse rates were established with functions from the R base package (*R Development Core Team, 2019*).

## Acknowledgements

This study acknowledges support by the ERC consolidator grant (no. 772852) led by Ulrike Herzschuh. We like to thank Sven Willner for assisting in programming and OpenMP improvements and Cathy Jenks for proofreading and improving the article. Further, we thank two reviewers and the topical handling editor for their comments which improved the article.

## Additional information

### Competing interests

The authors declare that no competing interests exist.

### Funding

| Funder | Grant reference number | Author |
|---|---|---|
| H2020 European Research Council | 772852 | Stefan Kruse Ulrike Herzschuh |

The funders had no role in study design, data collection and interpretation, or the decision to submit the work for publication.

### Author contributions

Stefan Kruse, Conceptualization, Data curation, Formal analysis, Investigation, Methodology, Project administration, Resources, Software, Validation, Visualization, Writing – original draft, Writing – review and editing; Ulrike Herzschuh, Conceptualization, Funding acquisition, Methodology, Project administration, Supervision, Visualization, Writing – original draft, Writing – review and editing

### Author ORCIDs

Stefan Kruse (iD) http://orcid.org/0000-0003-1107-1958

### Decision letter and Author response

Decision letter https://doi.org/10.7554/eLife.75163.sa1
Author response https://doi.org/10.7554/eLife.75163.sa2

## Additional files

### Supplementary files

• Transparent reporting form

### Data availability

We publicly uploaded the model code on Github https://github.com/StefanKruse/LAVESI/tree/crutransects (copy archived at swh:1:rev:422dc634a9d47e36c2e5c3a78906901ab6f2cdc3), and stored the commit used in this publication on Zenodo https://doi.org/10.5281/zenodo.6344261 Simulation results are stored on Zenodo https://doi.org/10.5281/zenodo.6484111.

The following datasets were generated:

| Author(s) | Year | Dataset title | Dataset URL | Database and Identifier |
|---|---|---|---|---|
| Kruse S | 2022 | Branch crutransects of LAVESI-WIND v1.2 | https://doi.org/10.5281/zenodo.6344261 | Zenodo, 10.5281/zenodo.6344261 |
| Kruse S, Herzschuh U | 2022 | Forest expansion for different warming scenarios simulated for 2010 to 3000 CE with LAVESI for Siberia | https://doi.org/10.5281/zenodo.6484111 | Zenodo, 10.5281/zenodo.6484111 |

The following previously published datasets were used:

| Author(s) | Year | Dataset title | Dataset URL | Database and Identifier |
|---|---|---|---|---|
| Arkhipov SM, Kotlyakov V, Punning YMK, Zogorodnov V, Nikolayev VI, Macheret YY, Zagorodnov VS, Barkov NI, Korsun SA, Korotkevich V, Morev VA, Evseyev AV, Vostokova TA, Andreev AA, Klementyev OL, Korotkevitch YS, Stiévenard M, Sinkevich SA, Samoylov OY, Vaikmaye R | 2008 | Deep drilling of glaciers: Russian projects 290 in the Arctic (1975-1995) | https://doi.org/10.1594/PANGAEA.707363 | PANGAEA, 10.1594/PANGAEA.707363 |
| Wieczorek M, Kruse S, Epp LS, Kolmogorov A, Nikolaev AN, Heinrich I, Jeltsch F, Pestryakova LA, Zibulski R, Herzschuh U | 2017 | Field data for larches growing in the Taimyr treeline ecotone | https://doi.org/10.1594/PANGAEA.874612 | PANGAEA, 10.1594/PANGAEA.874612 |

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

## Appendix 1

### Model performance

During simulation runs, each Environment grid cell of 20 × 20 cm needs 10 bytes, an element of the Tree structure 64 bytes, and the Seed structure 32 bytes. In total, for very dense scenarios it needs for each square kilometre ~11 GB RAM. The parallelization was updated and instead of Standard Template Library (STL) list containers for tree and seed elements a vector structure was implemented to allow better support for OpenMP (https://www.openmp.org) parallelization for scaling on a high-performance cluster and the realization of large-scale transect simulations as needed for this study. The computation speed depends on the number of trees and reaches $\sim 3.5\,\mathrm{s\,year^{-1}\,km^{-2}}$ for very dense forests on eight cores that can be reduced when using more cores (*Appendix 1—figure 1*). However, due to overheads, scaling is not 1:1. Of the internal functions of LAVESI, the mortality is most computationally intensive as each seed and tree need to be handled individually.

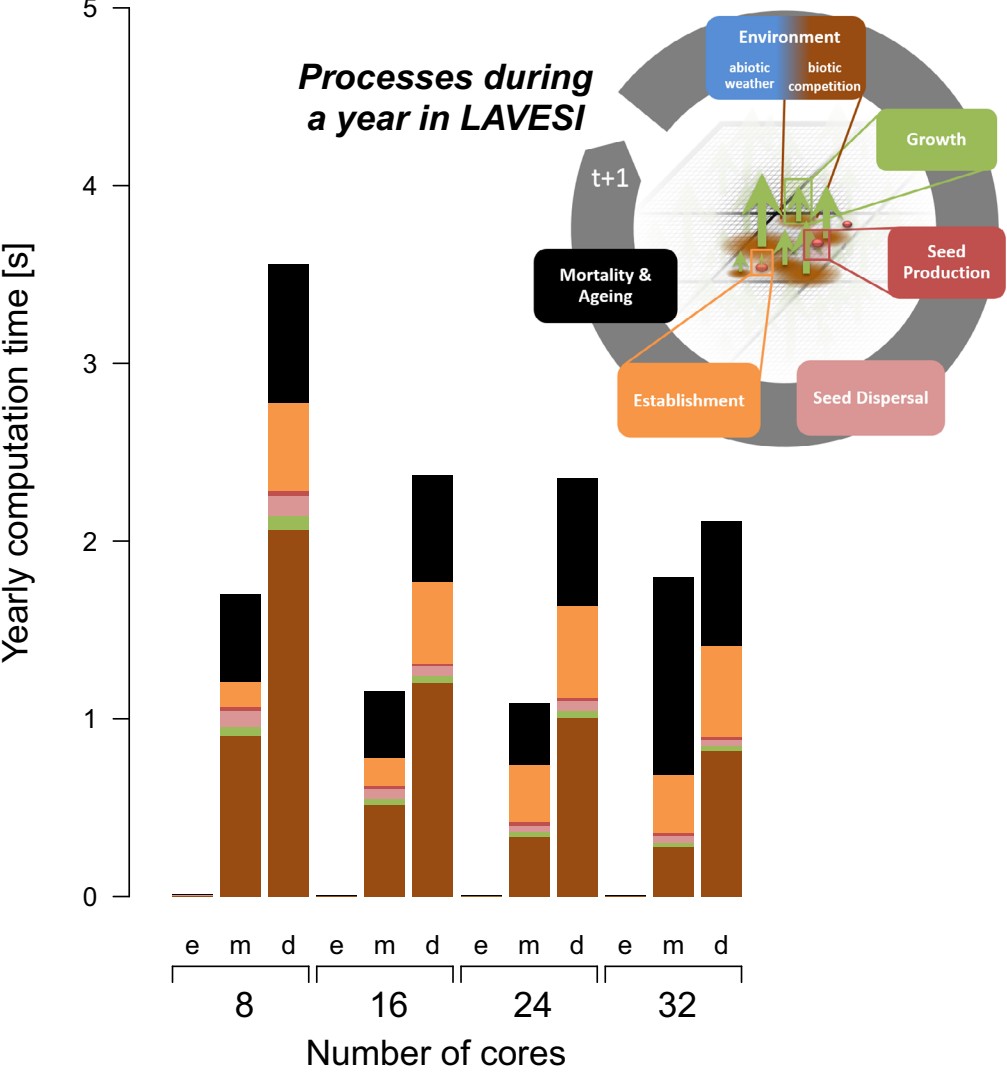

**Appendix 1—figure 1.** The computation speed increases with the number of available threads and reaches a plateau caused by overheads. The processes of LAVESI scale differently and the process Environment is computationally the heaviest and thus relatively faster with more cores while processes such as Mortality show no clear benefit of more threads. Data were aggregated for 100 years at three different population dynamics stages: e, empty; m, mature old growth; d, densification.

# Appendix 2

## Migration rates

*Appendix 2—table 1* provides a summary of simulated treeline advance and the corresponding position at the same year of the climate analogue. *Appendix 2—figure 1* shows the position of the climate-analogue treeline position at the four transects, and *Appendix 2—figure 2* gives an overview of the simulated maximum positions of the single-tree stands, treeline, and forestline at the same transects.

**Appendix 2—table 1.** Summary of simulated vs. climate envelope-based treeline advance.
Climate envelope solely based on July temperatures (following *MacDonald et al., 2007*). Values in bold reach the shoreline, but note that because of technical reasons the step sizes of the climate are in 10-km steps. Values are the result of three simulation repeats, and the standard deviation is stated next to the mean value.

| Transect | Scenario | Maximum potential expansion based on climate and year | Simulated maximum expansion and year | Treeline migration rate 2000–2100 km decade$^{-1}$ | Treeline migration rate 2000–2300 km decade$^{-1}$ |
|---|---|---|---|---|---|
| Taimyr | RCP 2.6* | 210 km 2090 | 192.2 ± 18.5 km 2733 ± 162 | 0.8 ± 0.7 | 1.5 ± 0.1 |
| | RCP 2.6*c | 210 km 2090 | 135.3 ± 1.4 km 2357 ± 58 | 3.4 ± 2.9 | 1.9 ± 0 |
| | RCP 2.6 | 320 km 2090 | 344.8 ± 2.9 km 2730 ± 87 | 11.5 ± 2.3 | 9 ± 0 |
| | RCP 2.6c | 320 km 2090 | 331.3 ± 1.4 km 2283 ± 29 | 8.6 ± 1.2 | 8.6 ± 0.1 |
| | RCP 4.5 | 410 km 2297 | **544.5 ± 0 km 2383 ± 6** | 23.4 ± 0.2 | 10.6 ± 0.1 |
| | RCP 4.5c | 400 km 2221 | 405.3 ± 3.8 km 2567 ± 231 | 24.7 ± 0.2 | 11.2 ± 0.1 |
| | RCP 8.5 | **560 km 2082** | **544.5 ± 0 km 2160 ± 0** | 29.1 ± 0.5 | 16.1 ± 0 |
| | RCP 8.5c | **560 km 2082** | **544.5 ± 0 km 2140 ± 0** | 28.4 ± 0.4 | 16.2 ± 0 |
| Buor Khaya | RCP 2.6* | **130 km 2040** | 94.5 ± 0.9 km 2307 ± 75 | 0.1 ± 0.2 | 0.2 ± 0 |
| | RCP 2.6*c | **130 km 2040** | 91 ± 0.9 km 2497 ± 318 | 0 ± 0 | 0 ± 0 |
| | RCP 2.6 | **130 km 2020** | 120.2 ± 1.4 km 2247 ± 12 | 1 ± 0.3 | 1.1 ± 0 |
| | RCP 2.6c | **130 km 2020** | 122.3 ± 0.3 km 2633 ± 92 | 0.9 ± 0.6 | 1.2 ± 0.2 |
| | RCP 4.5 | **140 km 2133** | 124.5 ± 0 km 2303±6 | 0.6 ± 0.5 | 1.3 ± 0.1 |
| | RCP 4.5c | **140 km 2133** | 124.5 ± 0 km 2287 ± 12 | 0.2 ± 0.1 | 1.2 ± 0.1 |
| | RCP 8.5 | **140 km 2038** | 124.5 ± 0 km 2120 ± 0 | 3.2 ± 0 | 1.2 ± 0 |
| | RCP 8.5c | **140 km 2038** | 124.5 ± 0 km 2123 ± 12 | 3 ± 0.3 | 1.1 ± 0 |
| Kolyma | RCP 2.6* | 130 km 2012 | 120.8 ± 0.3 km 2330 ± 0 | 1.2 ± 0.1 | 0.7 ± 0 |
| | RCP 2.6*c | 130 km 2012 | 119 ± 0.9 km 2217 ± 12 | 1.1 ± 0.2 | 0.7 ± 0 |

*Appendix 2—table 1 Continued on next page*

*Appendix 2—table 1 Continued*

| Transect | Scenario | Maximum potential expansion based on climate and year | Simulated maximum expansion and year | Treeline migration rate 2000–2100 km decade$^{-1}$ | Treeline migration rate 2000–2300 km decade$^{-1}$ |
|---|---|---|---|---|---|
| | RCP 2.6 | 150 km 2094 | 134.3 ± 0.3 km 2230 ± 0 | 2.1 ± 0.1 | 1.2 ± 0 |
| | RCP 2.6c | 150 km 2094 | 113.5 ± 17.3 km 2103 ± 92 | 1.3 ± 0.6 | 1 ± 0.1 |
| | RCP 4.5 | 150 km 2284 | 134.5 ± 0 km 2257 ± 12 | 2 ± 0.1 | 1.2 ± 0 |
| | RCP 4.5c | 140 km 2121 | 134.5 ± 0 km 2307 ± 6 | 1.6 ± 0.5 | 1.2 ± 0 |
| | RCP 8.5 | 150 km 2072 | 134.5 ± 0 km 2143 ± 6 | 2.2 ± 0.2 | 1.2 ± 0 |
| | RCP 8.5c | 150 km 2072 | 134.5 ± 0 km 2140 ± 0 | 2.2 ± 0.1 | 1.2 ± 0 |
| Chukotka | RCP 2.6* | 300 km 2047 | 285.2 ± 8.4 km 2173 ± 29 | 25.1 ± 0.1 | 8.8 ± 0.1 |
| | RCP 2.6*c | 300 km 2047 | 285.7 ± 3.2 km 2203 ± 29 | 25.3 ± 0.2 | 9.2 ± 0.1 |
| | RCP 2.6 | 350 km 2047 | 363.5 ± 0 km 2423 ± 6 | 30.8 ± 1 | 11.3 ± 0.2 |
| | RCP 2.6c | 350 km 2047 | 358.8 ± 6.4 km 2290 ± 87 | 30.9 ± 0.2 | 11.3 ± 0.2 |
| | RCP 4.5 | 380 km 2144 | 604.5 ± 0 km 2747 ± 29 | 27.2 ± 1.2 | 12 ± 0 |
| | RCP 4.5c | 380 km 2144 | 377 ± 0 km 2297 ± 6 | 28 ± 0.2 | 11.9 ± 0.1 |
| | RCP 8.5 | 610 km 2107 | 604.5 ± 0 km 2160 ± 0 | 29.1 ± 0.3 | 19.6 ± 0 |
| | RCP 8.5c | 610 km 2107 | 604.5 ± 0 km 2150 ± 0 | 29.2 ± 0.1 | 19.7 ± 0 |

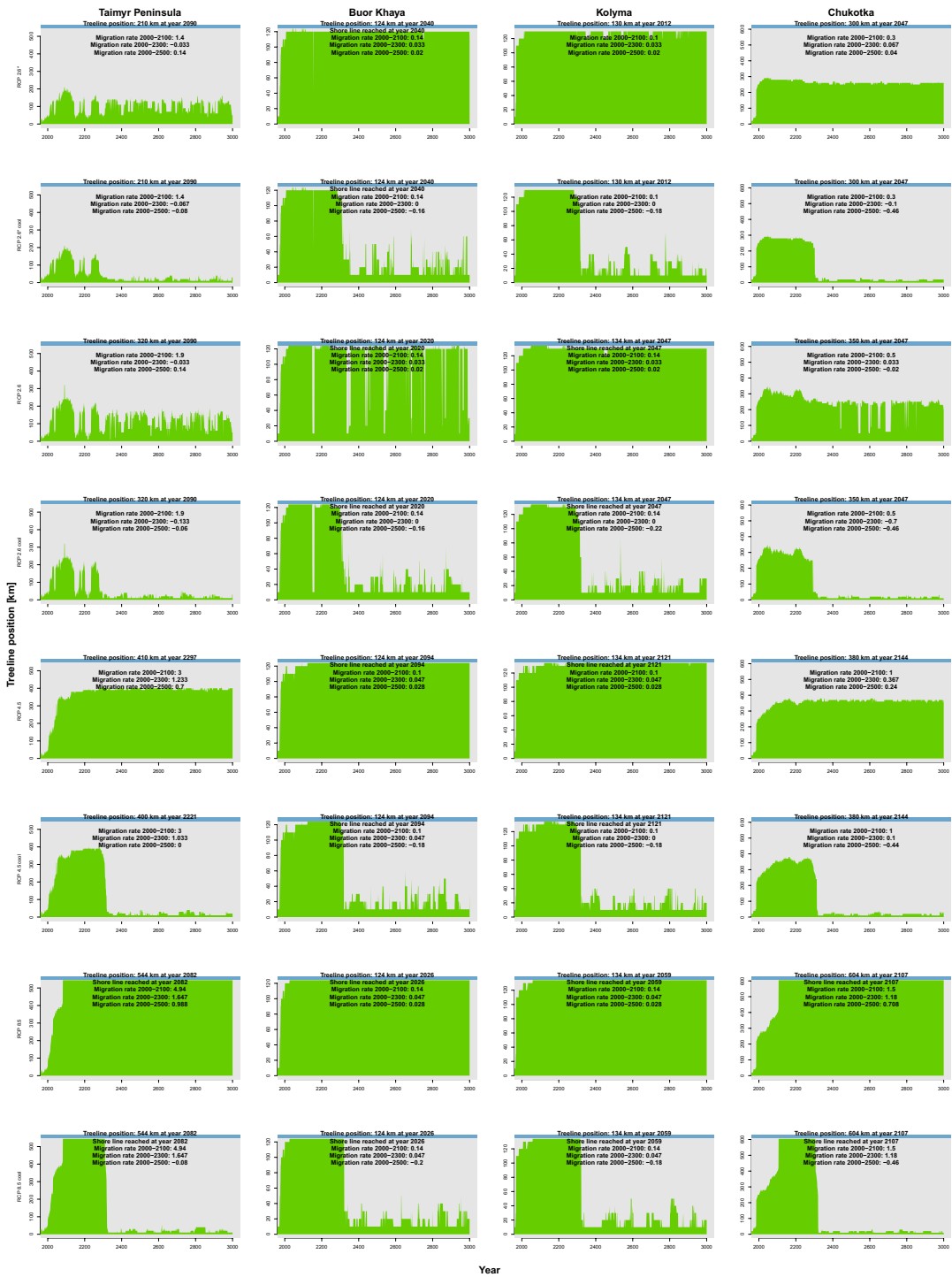

**Appendix 2—figure 1.** Climate-analogue treeline position along the four transects (columns) based on extrapolated July temperatures from the relative concentration pathway (RCP) scenarios (rows). A general rapid expansion of the treeline (continuous cover from south $\geq 1$ tree ha$^{-1}$, green) can be seen. In each panel, the maximum treeline position and the corresponding year are provided next to migration rates in three periods in km decade$^{-1}$. Blue colour at the top represents the shoreline.

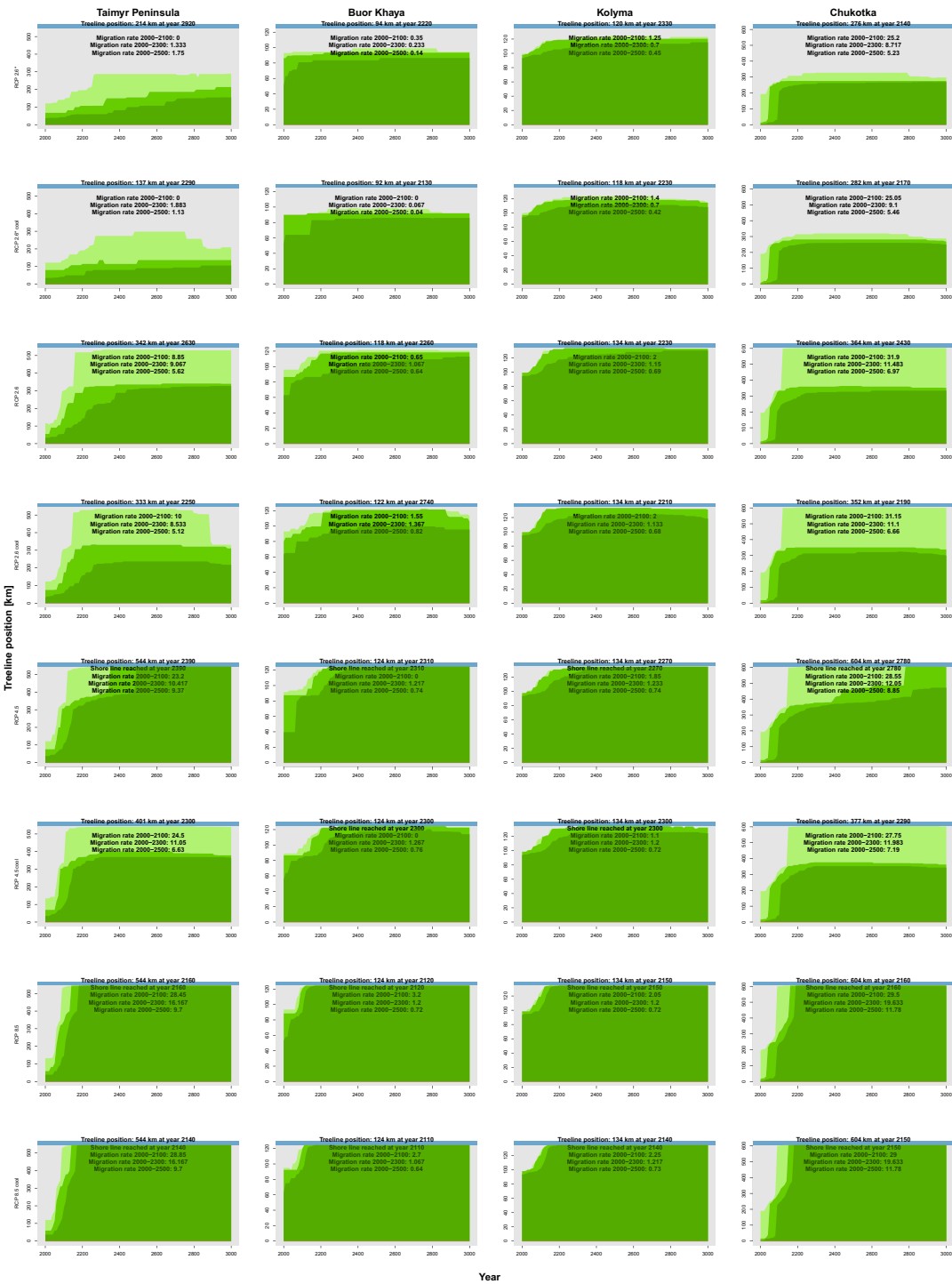

**Appendix 2—figure 2.** Simulated forest expansion dynamics along the four transects (columns) forced with climate data based on relative concentration pathway (RCP) scenarios (rows). A general quick expansion of single-tree stands (northernmost position of forest islands ahead of the treeline with $\geq 1$ stem ha$^{-1}$, light green) and the treeline (northernmost position of a continuous forest cover with $1$ stem ha$^{-1} \leq$ density $< 100$ stem ha$^{-1}$, green) followed by the forestline (northernmost position of a forest cover $> 100$ stems ha$^{-1}$, dark green) is seen. Blue colour at the top represents the shoreline.

## Appendix 3

### Treeline migration trajectories

*Appendix 3—figure 1* shows the treeline trajectories of the position based on simulations and the climate-analogue position at the four transects and the time when the simulated positions reach the first time the same position of the climate-analogue.

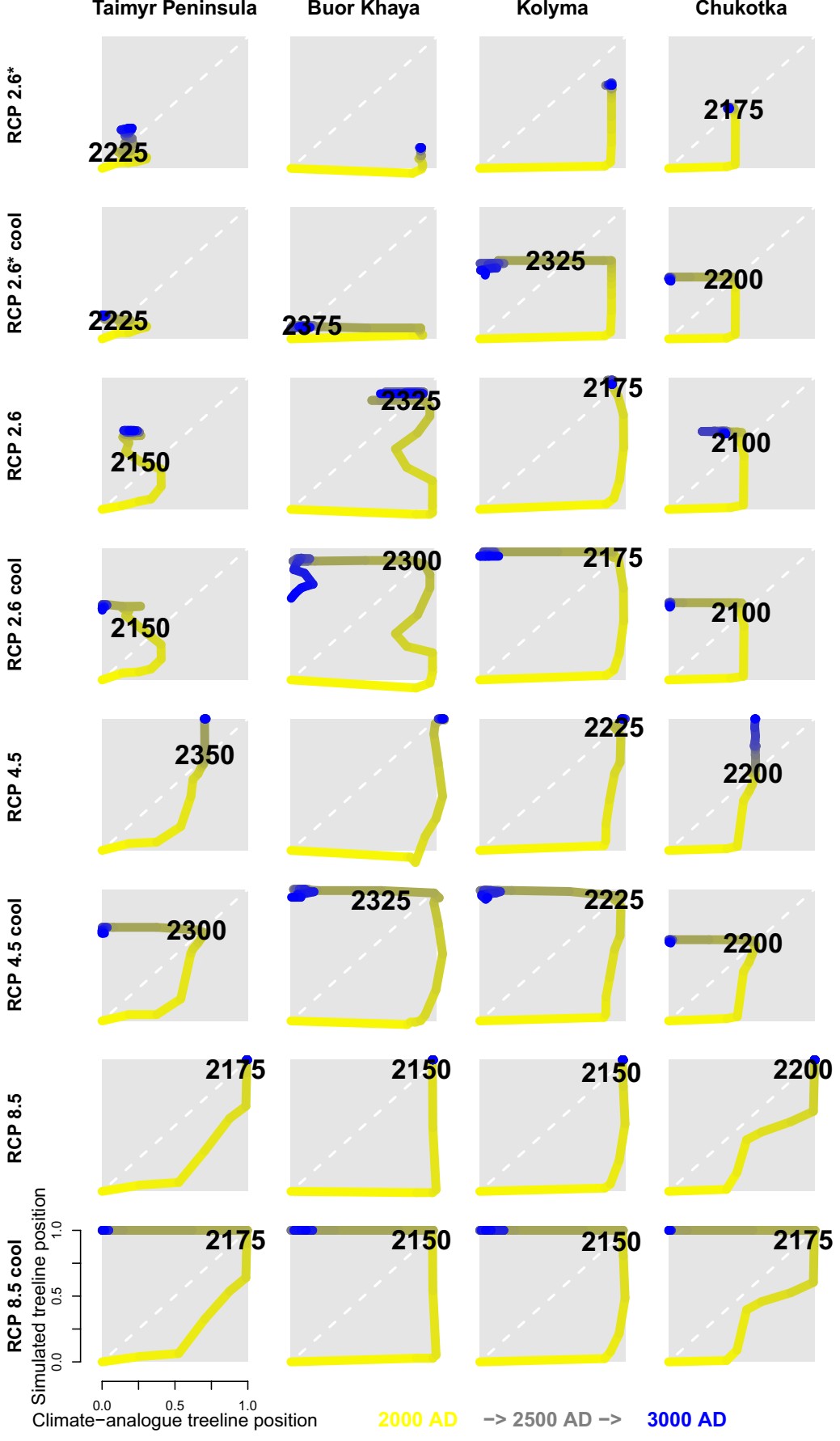

**Appendix 3—figure 1.** Trajectories for all four regions. Numbers are the first year when the simulated treeline position is equal to or farther north than the modern climate-analogue position. Colour of line segments ranges from yellow for year 2000 to blue in 300 CE.

## Appendix 4

### Tundra area dynamics

*Appendix 4—figure 1* gives an overview of the tundra area development in 100-year time steps for each of the eight climate scenarios.

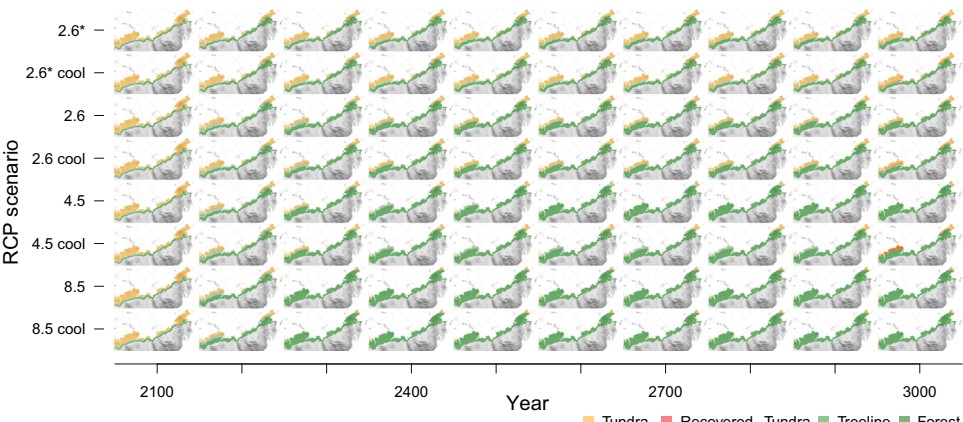

**Appendix 4—figure 1.** Forest expansion for 100-year time steps and all scenarios. Modern tundra areas will become covered by at least open larch tundra forests under different climate scenarios and nearly reach the shoreline in warmest conditions.

