## [Editor Report]

This study will be of interest to researchers studying climate change and Arctic tundra systems. The authors apply LAVESI, a machine-intensive and spatially explicit simulation of individual Siberian trees at the tundra-forest boundary, to call attention to the rapid reduction in the tundra biome as climate warming pushes forests toward the Arctic Ocean. This detailed modelling study predicts dramatic losses of tundra area by the middle of the millennium even under an ambitious climate mitigation scenario and highlights considerable risks of extinction.

---

## [Decision Letter]

**Decision letter after peer review:**

Thank you for submitting your article "Regional opportunities for Siberian tundra conservation in the next 1000 years" for consideration by *eLife*. Your article has been reviewed by 2 peer reviewers, and the evaluation has been overseen by a Reviewing Editor and Meredith Schuman as the Senior Editor. The following individuals involved in review of your submission have agreed to reveal their identity: Roman Dial (Reviewer #1).

Essential revisions:

1) Systematically discuss in the introduction or the appendix the main limiting factors of tree establishment and growth relevant for the study area, and mentioning those finally implemented in the model would add considerable value (i.e. limiting factors that prevent tree establishment and growth, permafrost degradation, soil nutrient development, biotic interactions (herbivory)).

This discussion would increase traceability of methods and assessment of relevance of results, but also further emphasize how much this study is an improvement over previous studies by including some of these processes largely neglected earlier. Some of this very relevant information is mentioned in the response letter, but only partly introduced in the revised manuscript.

2) Discussion of limitations of the modelling study is largely missing, including the following aspects:

– Is this vegetation model coupled with a climate model? If not, feedbacks of forest expansion with climate and permafrost are currently neglected. The model is tested along gradients in selected regions, but it remains uncertain if space-for-time approach will hold in the future and further north, esp. when large-scale feedbacks are included.

– What about disturbances and extreme weather conditions that might regionally impact treeline advance or tree survival? E.g. increasing tundra fire activity might strongly impact vegetation development. Also, droughts/flooding might lead to regional vegetation impacts, esp. at seedling stage. Extreme events and related disturbances are predicted to increase under climate change and a discussion on how they might impact predictions by the model is needed. Are these factors all only short–term and neglectable compared to the long–term perspective modelled? If yes, this should be mentioned.

3) The Short Report format is appropriate for this study but does entail space limitations. Please relegate some description of machine computations to the supplementary material, and use the space for a fuller description in the methods of how warming, wind, and water are included in model parameterization (rather than citing a previous paper behind a pay–wall). Related: Figure 3 is both a computational and a conceptual graphic. Many readers might prefer to understand how climate change is incorporated into LAVESI conceptually, at least as much if not more so than how much computational time is required to run it.

– Methods: "Climate forcing" and "Tree growth on transects" needs at least some connection beyond citation of a previous paper.

– Line 205+: "July temperatures (Figure 4), which have the strongest impact in the model" Through what mechanisms? It would help a field biologist or plant physiologist to understand and evaluate the model.

– Line 209+: "Wind speed and direction data" How are these used in the model? What role do they play? A naïve reader would assume they play a role in dispersal if from the south and a role in mortality if from the north.

– Lines 259–272: These could be reduced and simplified to make room for the more valuable information about how the model parameters and structure are structurally related to July temperature and winds in winter or summer and to precipitation. I realize that these topics are likely detailed in previous publications, but a brief review or sketch even if put in another appendix would be invaluable to evaluating and understanding this very interesting model.

4) Clean–up

– In general, figures and graphs need to be checked and improved; expand legend, indicate axes units consistently, some figures and their legend not matching in appendix, check references to figures and tables in text, etc.

– Figure 1. Is the upper left panel of Appendix 2 Figure 1 the same as this and so redundant? The sub–panel in Figure 1 is tantalizing but not clear how best to interpret. Mention what RCP2.6* is standing for. hard to interpret graph: 4 scenarios mentioned, but 7 lines criss–crossing? x– and y–axis units? very hard to separate 25 years segments, is entire time–range until year 3000 displayed? What are the units of the axes? Does 0 = current forest boundary and 1 = Arctic Ocean coast? add more detail to legend, so it is self–explanatory. Like the diagonal is where climate and treeline are in equilibrium; below the diagonal tree migration lags climate (as it always will when isotherms move faster than dispersal); and above the diagonal is "overshooting" when there is some sort of momentum in the movement of treeline? Also, maybe arrows rather than colors would more readily indicate the direction of time?

– Figure 2: annotate x–axis of graphs around maps with years. This figure is too small to be well readable.

– Figure 3 "Data were aggregated for 100 years at three different population dynamics stages: e=empty, m=mature old growth, d=densification." Relevance of this caption and figure is unclear (other than the conceptual graphic which is good but could be excellent with the addition of where temperature, the main driving variable, fits in) to the paper about forest migration and tundra disappearance?

– Figure 4 (map) coming first would help orient Nearctic researchers better to the frequent reference to regional geography. Hard to distinguish the different RCPs versus treeline, centre, shoreline due to similar line symbols (maybe just use colors for 3 different locations within transect or RCP scenarios instead of different W–E locations) (4 transect locations can be indicated in the graphs with their name, no need to color code this information). RCP legend: include dots (e.g. 2.6 instead of 26) and name RCP1.3 RCP2.6* for consistency with text.

– Check that Table 1 is provided: the link to "Table 1" goes to Appendix 1 Table 1 which Reviewer #1 struggled to read. Do we need so many decimal places? Here in Appendix 1 Table 1, what exactly are the {plus minus} referring to? Why two columns at the right margin of the table that appear to overlap in their years of coverage?

– Use consistent units (km/decade is likely more appropriate than m/year).

– L33/34: 'The Siberian tundra areas are known for their alpine species biodiversity' – unclear why the arctic–alpine species are mentioned specifically. The Siberian tundra area covered by this study is important as it shows a high regional floristic diversity, representing almost all arctic tundra types. E.g. lowland tundras with their vegetation are very important for migratory bird species, so it might be misleading to highlight the arctic–alpine species diversity only rather than emphasizing the regional floristic diversity.

– Lines 38–39: "the modelled climate envelope for tundra migrates north by ~ 290 *m yr*-1 (Loarie et al., 2009)." Perhaps a more accurate statement from that paper would read, "the climate velocity of isotherms within the tundra biome have been calculated as ~ 290 *m yr*-1" although in that paper the word "velocity" seems loosely applied and climate speed would seem more appropriate except when it's claimed a direction (like up in elevation or north in latitude).

– Lines 39–40: "When assuming a direct relationship linking the modern position of the treeline to its main limiting factor of July temperature…" Do you mean this "When assuming a direct equilibrial relationship linking the modern position of the treeline to its main limiting factor of July temperature…"? The paper would benefit from more consistent and integrated use of terms.

– Lines 46–49: "This and the fact that in Siberia the treeline is formed solely by larch

species growing on permafrost while in North America this niche is covered by a mix of species (Mamet et al., 2019; Herzschuh, 2020), make it impossible to transfer knowledge from any other treeline region to inform Siberian tundra responses." Suggested word change, because nowhere is a conifer treeline moving rapidly, right?

"This and the fact that in Siberia the treeline is formed solely by larch species growing on permafrost while in North America this niche is covered by a mix of species (Mamet et al., 2019; Herzschuh, 2020), make it difficult if not impossible to transfer knowledge from any other treeline region to inform Siberian tundra responses."

– Lines 50–52: "Equilibrium–based estimations are not backed by empirical studies which are showing slower responses and either no treeline advance or rates of only a few metres per year in Siberia (Scherrer et al., 2020; Kharuk et al., 2013; Wieczorek et al., 2017b; Harsch et al., 2009)." Suggest replacement of Hofgaard et al., 2012 with Rees et al. 2020 as most up to date review of treeline speeds. Rees, W.G., Hofgaard, A., Boudreau, S., Cairns, D.M., Harper, K., Mamet, S., Mathisen, I., Swirad, Z. and Tutubalina, O., 2020. Is subarctic forest advance able to keep pace with climate change? Global Change Biology, 26(7), pp.3965–3977.

– L54: why is the focus on intra–specific competition here? why not interspecific competition across taxa, including insect, mammal herbivory etc?

– Lines 80–81: Would the authors consider introducing their results with a more believable biotic velocity (if I am understanding "treeline advance follows climate warming" correctly) than 3 km y–1 (i.e. 30 km decade–1). That seems very unreasonable!

Also: check figure and table references throughout – e.g. should be Appendix Table 1, and on L112 (see below) Figure 2B cannot be identified.

– L112: see above – you refer to Figure 2b, but no Figure 2b.

– Line 251+: “single–tree stands” should these be “0 tree ha–1 < density {less than or equal to} 1 tree ha–1” because otherwise as written “[density] > 1 *stem* (tree > 1.3 *m* tall) per ha” would include the other two classes since both treeline and forest line have “densities > 1 stem per ha”. Mathematically, “forest cover not falling below 1 *stem* h*a*−1” is stated as “forest cover {greater than or equal to} 1 *stem* h*a*−1” and “forest cover not falling below 100 *stems* h*a*−1” is stated as “forest cover {greater than or equal to} 100 *stem* h*a*−1”. These confusing descriptions show up again in Appendix 1 Figure 2. Caption.

– Line 255+: Also unclear is this statement: "The determined treeline at year 2000 is used as a baseline for expansion and subtracted from the following years' values."

– Is "climate analogue" defined somewhere? Is the idea related to "climate velocity"?

– Is "simulated treeline position" related to the idea of "biotic velocity"?

– L 476: is appendix Figure 1 showing treeline position or as mentioned in legend forestline position?

– Appendix 1 Figure 1. Vs Appendix 1 Figure 2: It's not obvious why these are showing different patterns when the captions are so similar: Is Appendix 1 Figure 1. "climate velocity" sensu Loarie et al. (2008) And Appendix 1 Figure 2 "biotic velocity" sensu Ordonez, A. and Williams, J.W., 2013. Climatic and biotic velocities for woody taxa distributions over the last 16 000 years in eastern North America. Ecology Letters, 16(6), pp.773–781.

– Appendix 1 Figure 1:"Forest expansion dynamics along the four transects (columns) extrapolated based on July temperatures (climate–analogue) for climate data based on RCP scenarios (rows). What do the negative values represent in migration rates? And why are the rates averaged beginning with 2000 instead of averaged over the particular intervals? Intervals would seem more informative: 2000–2100, 2100–2300, 2000–2500. indicate unit of y–axis (or provide/explain in legend) and unit of migration rate. Treeline position in each panel – is this the northernmost treeline position identified and the corresponding year?

– Appendix 1 Figure 2: "Forest expansion dynamics along the four transects (columns) forced with climate data based on RCP scenarios (rows)." is green not treeline and dark green forestline? These definitions are quite hidden and come very late on line 251 while they are very important!

– Appendix 2 Figure 1. Needs some text labeling each figure by transect. "Numbers are the first year when the simulated treeline position is equal to or farther north than the modern climate–analogue position." I see no numbers other than the RCP scenario identifiers. No years indicated, no blue color, seems to be different graph. Should also be moved to Appendix 2 section.

– Appendix 2 Figure 2. The caption references Figure 2 but that seems an error. If it is not, please tell the reader what exactly they should look for in Figure 2. no red in figure mentioned in legend.

– Finally, some references to other individual based models like Rupp, T.S., Chapin, F.S. and Starfield, A.M., 2001. Modeling the influence of topographic barriers on treeline advance at the forest–tundra ecotone in northwestern Alaska. Climatic change, 48(2), pp.399–416.

---

## [Author Response]

Essential revisions:1) Systematically discuss in the introduction or the appendix the main limiting factors of tree establishment and growth relevant for the study area, and mentioning those finally implemented in the model would add considerable value (i.e. limiting factors that prevent tree establishment and growth, permafrost degradation, soil nutrient development, biotic interactions (herbivory)).This discussion would increase traceability of methods and assessment of relevance of results, but also further emphasize how much this study is an improvement over previous studies by including some of these processes largely neglected earlier. Some of this very relevant information is mentioned in the response letter, but only partly introduced in the revised manuscript.

Following the recommendations of the reviewer, we have included new paragraphs with a more detailed presentation of the implemented abiotic/biotic limiting factors in the Methods section. This includes either a statement of how these are explicitly considered, or argumentation as to whether these are implicitly considered and thus already part of the model LAVESI.

“Many factors influence the success of tree establishment and growth, of which the following seem most important to understand treeline migration. Warmth, especially summer mean temperatures, cumulative temperatures, and the vegetation period length, are key climate variables (e.g. MacDonald et al., 2008) and are hence included in the model for estimating potential tree growth and tree mortality (Kruse et al., 2016). This is further influenced by the abiotic (permafrost, active-layer depth) and biotic (intraspecific competition) environment. For example, permafrost soils with a shallow active-layer depth limit the space available for rooting and nutrient stores (e.g. Sullivan et al., 2015) and frozen/cold soils prevent growth but water can support growth in dry summers (Sugimoto et al., 2002). The latter functioning could be lost in future hot environments and in consequence it would decrease the treeline advance rate further and thereby increase the lag time. Accordingly, the active-layer depth is explicitly simulated in the model’s environment module, but soil nutrients are considered only implicitly. Individuals compete for space which serves as a surrogate for this. Permafrost degradation/abrupt thaw and warming (Biskaborn et al., 2019; Stuenzi et al., 2021) could lead to thermokarst processes that would locally cause swamping/waterlogging, leading to suppressed establishment/growth (Rees et al., 2020), which is not explicitly simulated and unlikely to limit tree invasion on a broader scale. Further limitations could arise from the lack of a suitable mycorrhiza symbiosis partner, but as shown by Hippel et al. (2021) even in northernmost areas on the Taimyr Peninsula mycorrhiza are present even before the invasion of trees in the Holocene. Additionally, dispersal rates of mycorrhiza with small spores are a magnitude faster than for taxa with larger seeds so that geographic distance is unlikely to be a limiting factor, similar to diatoms as shown by Stoof-Leichsenring et al. (2015). On the other hand, interspecific competition of recruits with present shrubs or grasses which are increasing more quickly – called the greening of the Arctic (Pearson et al., 2013; Berner et al., 2020) – is not explicitly simulated in LAVESI, although it is partly included in the germination probability and survival rate of individuals at the seedling stage. Further biotic interactions are implicitly included, such as herbivory by browsers (e.g. Wielgolaski et al., 2017), which reduces the survival and growth rate by damaging the shoots (compare krummholz study by Kruse et al., 2020) and can lead to a snow compaction cooling effect on soil temperatures (Beer et al., 2020), or pest outbreaks that are likely to increase in the future (e.g. larch silk moth, Fedotova, 2019). This would most likely lead to a further reduction of migration rates.

2) Discussion of limitations of the modelling study is largely missing, including the following aspects:– Is this vegetation model coupled with a climate model? If not, feedbacks of forest expansion with climate and permafrost are currently neglected. The model is tested along gradients in selected regions, but it remains uncertain if space-for-time approach will hold in the future and further north, esp. when large-scale feedbacks are included.

We did not couple LAVESI to a climate model but as stated used only climate forcing, thus allowing the model to be run in stand-alone mode with data from different sources. Nonetheless, as presented in Pearson et al. (2013) tundra conversion to forests will ultimately lead to further warming feedback and hence a faster rate of conversion could be expected. We added a paragraph at the end of the Methods subsection ‘Tree growth on transects’: “The model used for this study is solely forced by climate data and no vegetation feedback on permafrost soils and climate was implemented. Therefore, the expected decrease of the albedo leading to a positive feedback with climate when tundra is colonized by tree stands (e.g. Pearson et al., 2013) is neglected here. Considering this, would likely further increase the rate of transition from tundra to forests.”

Regarding the space-for-time approach, we included a clarification and short discussion at the end of the Model section ‘Model tuning and validation’ we added the text: “We assume that the fundamental drivers of tree growth remain stable in time. The space-for-time approach used here is based on using several sites from across the treeline in the regions of interest for model parameterization and validation that have experienced a range of past climate (Pickett, 1989). However, climate warming will lead to unprecedented temperatures (Figure 1) that will affect the current stand structure and distribution. This may cause changes in the response of single considered processes or make other processes that are not currently important necessary (e.g. Blois et al., 2013; Pickett, 1989). Keeping this in mind, the future simulations should be interpreted with caution, although a simulation study with LAVESI revealed realistically predicted migration rates and patterns for the recent decades of strong warming in the study region of Chukotka (currently in review).”

– What about disturbances and extreme weather conditions that might regionally impact treeline advance or tree survival? E.g. increasing tundra fire activity might strongly impact vegetation development. Also, droughts/flooding might lead to regional vegetation impacts, esp. at seedling stage. Extreme events and related disturbances are predicted to increase under climate change and a discussion on how they might impact predictions by the model is needed. Are these factors all only short–term and neglectable compared to the long–term perspective modelled? If yes, this should be mentioned.

The reviewer brings attention to an important point. In our model, currently (tundra) fire is only implicitly considered, however drought impact on growth and survival is considered explicitly. On the other hand, flooding and waterlogging, that may take place locally, are not. The impact on predictions could be that the tundra colonization is even slower, hence, further prolonging the time-lagged response of the treeline migration. In the long run, an increased number of extreme weather events could support a faster dieback and supports a faster retreat of the treeline, which seems to overshoot and not return as quickly to its climate-analogue position. We added a paragraph at the end of the Methods section ‘Climate forcing’: “Future climate changes will lead to an increase in extreme weather events (Masson-Delmotte et al., 2021), among others a higher ignition probability and thus wildfires are expected to which the Siberian boreal forests are sensitive (e.g. Shvidenko and Schepaschenko, 2013). Fire is not yet explicitly simulated in the model and in a hotter environment fires could lead to dramatic forest losses. Areas might be turned into steppe and not revert to the prior tundra. Increases in extreme rainfall events will lead to local flooding and waterlogging (Masson-Delmotte et al., 2021), a response not explicitly implemented in the model. The anticipated effect to the LAVESI predictions is that tundra colonization would most likely be slowed down in the lowlands but less so in the well-drained mountainous regions, which would further prolong the time-lagged response of the treeline migration.”

3) The Short Report format is appropriate for this study but does entail space limitations. Please relegate some description of machine computations to the supplementary material, and use the space for a fuller description in the methods of how warming, wind, and water are included in model parameterization (rather than citing a previous paper behind a pay–wall). Related: Figure 3 is both a computational and a conceptual graphic. Many readers might prefer to understand how climate change is incorporated into LAVESI conceptually, at least as much if not more so than how much computational time is required to run it.– Methods: "Climate forcing" and "Tree growth on transects" needs at least some connection beyond citation of a previous paper.

We added further relevant details on the tree growth estimation based on climate forcing data in the Methods section’s first subsection ‘Model description’ – please see the above comment for a detailed text addition and below for minor changes at relevant positions in the text.

– Line 205+: "July temperatures (Figure 4), which have the strongest impact in the model" Through what mechanisms? It would help a field biologist or plant physiologist to understand and evaluate the model.

Here the impact on tree growth is meant. In short, warmer summer temperatures lead to a potential larger tree growth (growth is positively correlated with temperature and the vegetation period length), however, this is constrained by competition. This was added to the Methods section in the first subsection ‘Model description’.

For clarification we adjusted the text at the reviewer comment’s position to “July temperatures (Figure 1), which have the strongest impact in the model due to a positive influence on tree diameter growth, increase in these scenarios …”

– Line 209+: "Wind speed and direction data" How are these used in the model? What role do they play? A naïve reader would assume they play a role in dispersal if from the south and a role in mortality if from the north.

For clarifying the use of wind data we included further details in the Methods section in the first subsection ‘Model description’. See changes made above.

– Lines 259–272: These could be reduced and simplified to make room for the more valuable information about how the model parameters and structure are structurally related to July temperature and winds in winter or summer and to precipitation. I realize that these topics are likely detailed in previous publications, but a brief review or sketch even if put in another appendix would be invaluable to evaluating and understanding this very interesting model.

We thank the reviewer for the critical assessment of our manuscript. Following their suggestions, we removed the model performance plots from figure 3 and merged them, together with the text on lines 162-172, into the appendix and refer to it in the updated text of the manuscript.

Further, we included more details about how the forcing climate data link to the internal processes in the Methods section in the subsection ‘Model description’ after the first paragraph of a general description. Our changes also include a new conceptual graphic in figure 4 that shows the relevant processes of LAVESI and how climate warming will impact the individual processes in LAVESI.

The new text with a detailed model description is: “The simulation proceeds in yearly time steps from the beginning to the end of the input climate time-series following a stabilization period to ensure that emerging populations reach equilibrium with the environment. Stochasticity in the model was introduced by using random numbers generated with a pseudo random number generator. Within one simulation year, the following processes become consecutively invoked Figure 4: Update of environment: Interactions between neighbouring trees are local and indirect. Basal diameters of each individual tree are used to evaluate the competition strength. We use a yearly updated density map to pass information about competition for resources between trees. An active-layer depth is estimated based on an edaphic factor and the number of days exceeding 0 °C following simplifications by Hinkel and Nicholas (1995). Growth: The maximum possible growth is estimated based on 10-yearsmean climate auxiliary variables (temperature of the coldest (January) and warmest (July) month, *NDD*0 ‘net degree days’, *AAT*10 ‘active air temperature’). The individual growth of basal diameter and, if a tree reached a height of 1.3 m, of breast height diameter, is calculated from the maximum possible growth in the current year affected by the tree’s density index. From the resulting diameters, the tree height is estimated. Seed dispersal: Seeds in ‘cones’ are dispersed from the parent trees. The dispersal directions and distances are randomly determined from a ballistic flight influenced by wind speed and direction with decreasing probabilities for long distances and, if dispersed seeds leave the extent of the simulated plot, they can be introduced from the other side only on the east-west margins. Seed production: Trees produce seeds after the year at which they reached their stochastically estimated maturation height. The total amount depends on weather, competition, and tree size. Establishment: The seeds that lie on the ground germinate at a rate depending on current weather conditions. Mortality: Individual trees or seeds die, i.e. they become removed from the plot, at a specified mortality rate. For trees this is deduced from long-term mean weather values, a drought index, surrounding tree density, tree age and size, plus a background mortality rate. Seeds on the other hand have the same constant mortality rate whether on trees or the ground. Ageing: Finally, the age of seeds and trees increases once a year and seeds are removed from the system when they reach a defined species age limit.”

Figure 4 was newly added:

4) Clean–up– In general, figures and graphs need to be checked and improved; expand legend, indicate axes units consistently, some figures and their legend not matching in appendix, check references to figures and tables in text, etc.

We have edited and improved the figures in the manuscript. Please find our specific responses to the comments below.

– Figure 1. Is the upper left panel of Appendix 2 Figure 1 the same as this and so redundant? The sub–panel in Figure 1 is tantalizing but not clear how best to interpret. Mention what RCP2.6* is standing for. hard to interpret graph: 4 scenarios mentioned, but 7 lines criss–crossing? x– and y–axis units? very hard to separate 25 years segments, is entire time–range until year 3000 displayed? What are the units of the axes? Does 0 = current forest boundary and 1 = Arctic Ocean coast? add more detail to legend, so it is self–explanatory. Like the diagonal is where climate and treeline are in equilibrium; below the diagonal tree migration lags climate (as it always will when isotherms move faster than dispersal); and above the diagonal is "overshooting" when there is some sort of momentum in the movement of treeline? Also, maybe arrows rather than colors would more readily indicate the direction of time?

The figure presented in figure 1 is indeed the upper left panel in Appendix 2 Figure 1. We moved all four plots to Figure 2 in the main text and reduced the subpanel. Further, as requested, we added more detail to the caption. The figures show 8 individual paths, for each RCP scenario and the cooling and normal forcing, which was added for clarification in the caption as well. Additionally, we tested using arrows to indicate the direction and segment length and decided that this improves the figure further, which shows the full 3000 year range.

The caption was changed to: “The trajectory of the simulated treeline position relative to its current position and the maximum at the shoreline versus its climate-analogue position for the four regions shows a migration lag of the treeline during the first centuries (each line segment represents 25 years and the length of the arrow head corresponds to the step length) until the simulated treeline is limited by climate. Forests expand their area further and infilling proceeds when climate conditions cool and even overshoot in the long run with cooling back to 20th century temperatures. The diagonal is where climate and the treeline are in equilibrium; below the diagonal, tree migration lags climate; above the diagonal is "overshooting" and reaching locations actual climate would allow. For each RCP scenario two are presented, one for the scenario as-is and the second for the cooling; Scenario RCP 2.6* warms only at half the rate of RCP 2.6. See also Appendix 3 Figure 1 for the year when the trajectory passes the equilibrium.”

– Figure 2: annotate x–axis of graphs around maps with years. This figure is too small to be well readable.

We improved the figure by adding x- and y-axis labels to all graphs of tundra area dynamics and by increasing the size of the figure.

– Figure 3 "Data were aggregated for 100 years at three different population dynamics stages: e=empty, m=mature old growth, d=densification." Relevance of this caption and figure is unclear (other than the conceptual graphic which is good but could be excellent with the addition of where temperature, the main driving variable, fits in) to the paper about forest migration and tundra disappearance?

We removed this figure and its corresponding text from the main text and brought it to the Appendix as explained above in response to Essential Revisions 3.

– Figure 4 (map) coming first would help orient Nearctic researchers better to the frequent reference to regional geography. Hard to distinguish the different RCPs versus treeline, centre, shoreline due to similar line symbols (maybe just use colors for 3 different locations within transect or RCP scenarios instead of different W–E locations) (4 transect locations can be indicated in the graphs with their name, no need to color code this information). RCP legend: include dots (e.g. 2.6 instead of 26) and name RCP1.3 RCP2.6* for consistency with text.

We moved Figure 4 to the beginning and refer to it in the introduction. Further, we adjusted the colour to the position on the transect and added in the title the region name; changed RCP legend names, and corrected the typo in the caption.

– Check that Table 1 is provided: the link to "Table 1" goes to Appendix 1 Table 1 which Reviewer #1 struggled to read. Do we need so many decimal places? Here in Appendix 1 Table 1, what exactly are the {plus minus} referring to? Why two columns at the right margin of the table that appear to overlap in their years of coverage?– Use consistent units (km/decade is likely more appropriate than m/year).

Table 1 is actually located in the appendix. Unfortunately, we wrongly referred to this in the text, which is now corrected. We left the numbers with one decimal place if not referring to years, which we rounded. The plus/minus refers to the standard deviation of the results that stem from three simulation repeats. The two different periods over which the mean migration rate was calculated are to compare those to each other.

– L33/34: 'The Siberian tundra areas are known for their alpine species biodiversity' – unclear why the arctic–alpine species are mentioned specifically. The Siberian tundra area covered by this study is important as it shows a high regional floristic diversity, representing almost all arctic tundra types. E.g. lowland tundras with their vegetation are very important for migratory bird species, so it might be misleading to highlight the arctic–alpine species diversity only rather than emphasizing the regional floristic diversity.

We changed the text to mention also the high regional floristic diversity and added a citation to the corresponding text in the introduction.

The text is now: “The Siberian tundra areas are known for their high regional floristic diversity and species biodiversity (Pauli et al., 2012; Mod and Luoto, 2016; Arctic Climate Impact Assessment, 2004; Schmidt et al., 2017) and support special (indigenous) types of land use (e.g. reindeer herding, foraging).”

– Lines 38–39: "the modelled climate envelope for tundra migrates north by ~ 290 m yr−1 (Loarie et al., 2009)." Perhaps a more accurate statement from that paper would read, "the climate velocity of isotherms within the tundra biome have been calculated as ~ 290 m yr−1 " although in that paper the word "velocity" seems loosely applied and climate speed would seem more appropriate except when it's claimed a direction (like up in elevation or north in latitude).

“In Siberia, the climate velocity of isotherms within the tundra biome has been calculated as ~ 290 *m yr*−1 (Loarie et al., 2009).”

– Lines 39–40: "When assuming a direct relationship linking the modern position of the treeline to its main limiting factor of July temperature…" Do you mean this "When assuming a direct equilibrial relationship linking the modern position of the treeline to its main limiting factor of July temperature…"? The paper would benefit from more consistent and integrated use of terms.

Following your recommendation, we changed the sentence here. Furthermore, for a more consistent use of the terms in the paper, we updated the corresponding statements in the Results and Discussion sections first part in the beginning. Also, in the caption of Figure 2 (trajectories) we added that the diagonal means that the treeline position is in equilibrium with climate, see comment above.

Text here is now “When assuming a direct equilibrial relationship linking the modern position of the treeline to its main limiting factor of July temperature (MacDonald et al., 2008), the treeline is expected to reach its maximum expansion within the first centuries of the millennium.”

Text in Result and Discussion is now: “The simulations revealed that the treeline is not in an equilibrial relationship but its advance follows climate warming, potentially reaching 30 *km decade*−1 in the 21st century under warmest climate scenarios (Appendix 2 Table 1).”

– Lines 46–49: "This and the fact that in Siberia the treeline is formed solely by larchspecies growing on permafrost while in North America this niche is covered by a mix of species (Mamet et al., 2019; Herzschuh, 2020), make it impossible to transfer knowledge from any other treeline region to inform Siberian tundra responses." Suggested word change, because nowhere is a conifer treeline moving rapidly, right?"This and the fact that in Siberia the treeline is formed solely by larch species growing on permafrost while in North America this niche is covered by a mix of species (Mamet et al., 2019; Herzschuh, 2020), make it difficult if not impossible to transfer knowledge from any other treeline region to inform Siberian tundra responses."

We follow the statement of the reviewer that current observations show only slow migrations of treelines worldwide. However, we demonstrate that they potentially pick up speed, even attaining exponential rates. Accordingly, we adjusted the sentence as suggested.

– Lines 50–52: "Equilibrium–based estimations are not backed by empirical studies which are showing slower responses and either no treeline advance or rates of only a few metres per year in Siberia (Scherrer et al., 2020; Kharuk et al., 2013; Wieczorek et al., 2017b; Harsch et al., 2009)." Suggest replacement of Hofgaard et al., 2012 with Rees et al. 2020 as most up to date review of treeline speeds. Rees, W.G., Hofgaard, A., Boudreau, S., Cairns, D.M., Harper, K., Mamet, S., Mathisen, I., Swirad, Z. and Tutubalina, O., 2020. Is subarctic forest advance able to keep pace with climate change? Global Change Biology, 26(7), pp.3965–3977.

We understand that the reviewer meant Harsch 2009 here and in the following sentence. We replaced it by the suggested publication that was already cited elsewhere in the manuscript.

– L54: why is the focus on intra–specific competition here? why not interspecific competition across taxa, including insect, mammal herbivory etc?

We added to the Methods a section (see comment above) covering further interspecific competition and disturbances affecting tree taxa dynamics by e.g. pest outbreaks and herbivory.

Additionally, we changed the text in Discussion section/Line 86ff to:

“Other processes, not yet explicitly captured by the model, may constrain the migration process, such as the strong greening of the Arctic increasing interspecific competition (compare Pearson et al., 2013; Berner et al., 2020), the warming and abrupt thaw of permafrost (e.g. Stuenzi et al., 2021; Biskaborn et al., 2019), soil nutrients (e.g. Sullivan et al., 2015), and animal activity (e.g. Wielgolaski et al., 2017).”

– Lines 80–81: Would the authors consider introducing their results with a more believable biotic velocity (if I am understanding "treeline advance follows climate warming" correctly) than 3 km y–1 (i.e. 30 km decade–1). That seems very unreasonable!

It seems counterintuitive at first that rates of 3 km yr-1 are realistic knowing the debate about post-glacial migration (e.g. Feardeau 2013). However, rates of 500 m yr-1 have likely occurred. Knowing that small seeds have a good long-distance dispersal ability in this landscape as well as forming nuclei ahead of the treeline will enhance the speed of colonization (e.g. Kruse et al. 2019) such that these high rates may be realistic. In our study they were only obtained for the very hot scenarios, which have higher growth rates and the production of more seeds, both factors that positively influence the migration rate. We rephrased the first misleading sentence that could be read that rates even higher than 3 km yr-1 may be simulated and added a reference and a short debate about this in the Results and Discussion section.

The text there is now: “The simulations revealed that the treeline is not in an equilibrial relationship but its advance follows climate warming, potentially reaching 30 *km decade*−1 in the 21st century under warmest climate scenarios (Appendix 2 Table 1).”

In the end of the paragraph: “the simulated rates are high compared to those during post-glacial migration (e.g. Feurdean et al., 2013) but seem likely as future climate is unprecedented, especially in their rate of change.”

Also: check figure and table references throughout – e.g. should be Appendix Table 1, and on L112 (see below) Figure 2B cannot be identified.– L112: see above – you refer to Figure 2b, but no Figure 2b.

We checked the reference to the table, which as stated above was misleading. Figure 2b was not linked in the document and is now updated to refer to Figure 3 lower panel.

– Line 251+: “single–tree stands” should these be “0 tree ha–1 < density {less than or equal to} 1 tree ha–1” because otherwise as written “[density] > 1 stem (tree > 1.3 m tall) per ha” would include the other two classes since both treeline and forest line have “densities > 1 stem per ha”. Mathematically, “forest cover not falling below 1 stem ha−1” is stated as “forest cover {greater than or equal to} 1 stem ha−1” and “forest cover not falling below 100 stems ha−1” is stated as “forest cover {greater than or equal to} 100 stem ha−1”. These confusing descriptions show up again in Appendix 1 Figure 2. Caption.

The definitions at that location were not correct as information was missing about continuous tree cover as a prerequisite for the variables treeline and forestline. Hence, we edited the text for clarification to: “The positions of three key variables of stand densities expressed in stems, which are trees > 1.3 *m* tall, are extracted in 10-year steps: single-tree stands, defined as the northernmost position of forest islands ahead of the treeline with ≥ 1 *stem ha*−1; treeline stands, the northernmost position of a continuous forest cover with 1 *stem ha*−1 ≤ *density* 100 *stem ha*−1; and forestline stands, the northernmost position of forest cover > 100 *stems ha*−1 (see Figure 2 in Kruse et al., 2019a, for a graphical representation).”

And we also corrected the definitions in the captions of Appendix 1 Figures 1 and 2.

– Line 255+: Also unclear is this statement: "The determined treeline at year 2000 is used as a baseline for expansion and subtracted from the following years' values."

We clarified the statement to: “The position of the simulated treeline determined for year 2000 is used as the baseline for the calculation of migration rates.”

– Is "climate analogue" defined somewhere? Is the idea related to "climate velocity"?– Is "simulated treeline position" related to the idea of "biotic velocity"?

We use it in the results and discussion first paragraph “When relating and tracking the position of the treeline to the climate (=climate-analogue) …” but missed a proper definition in the Methods section. This was added to the subsection Simulation setup and data processing: “For comparison, we extrapolated the position of the modern observed treeline based on July temperatures (=climate-analogue) for years 2000-3000 CE. Therefore, the current July temperature at the treeline position is determined and tracked along the transect at each time step of the climate forcing.”

About the “simulated treeline position”; Yes, this key variable and its migration rate is the biotic velocity, similar to the climate velocity.

– L 476: is appendix Figure 1 showing treeline position or as mentioned in legend forestline position?

Corrected as stated above.

– Appendix 1 Figure 1. Vs Appendix 1 Figure 2: It's not obvious why these are showing different patterns when the captions are so similar: Is Appendix 1 Figure 1. "climate velocity" sensu Loarie et al. (2008) And Appendix 1 Figure 2 "biotic velocity" sensu Ordonez, A. and Williams, J.W., 2013. Climatic and biotic velocities for woody taxa distributions over the last 16 000 years in eastern North America. Ecology Letters, 16(6), pp.773–781.

It is not obvious from the caption that Appendix 1 (now Appendix 2) Figure 2 shows the simulated forest expansion dynamics, so we adjusted the caption for clarification to “Simulated forest expansion dynamics along the four transects (columns) forced with climate data based on RCP scenarios (rows). A general quick expansion of single-tree stands (northernmost position of forest islands ahead of the treeline with ≥ 1 *stem ha*−1, light green) and the treeline (northernmost position of a continuous forest cover with 1 *stem ha*−1 ≤ *density* < 100 *stem ha*−1, green) followed by the forestline (northernmost position of a forest cover > 100 *stems ha*−1, dark green) is seen. Blue colour at the top represents the shoreline.” And also the caption of Appendix 1 (now Appendix 2) Figure 1 was changed to “Climate-analogue treeline position along the four transects (columns) based on extrapolated July temperatures from the RCP scenarios (rows). A general rapid expansion of the treeline (continuous cover from south ≥ 1 *tree ha*−1, green) can be seen. In each panel the maximum treeline position and the corresponding year is provided next to migration rates in three periods in *km decade*−1. Blue colour at the top represents the shoreline.”

We agree with the reviewer that the migration rates we present are velocities. The velocities along the transects were estimated both for the climate-analogue treeline position (see above comments) and extracted from the simulations. Accordingly, we cite the studies suggested by the reviewer and include for clarity those terms adjusted to the text elsewhere, as mentioned in above comments and in the methods section to “The resulting values for each of the three key variables are used in 10-year steps to interpolate with a weighted average the expansion starting at the current treeline (e.g. Walker et al., 2005). This is similar to the biotic velocity of Ordonez and Williams (2013). Opposed to this, we extrapolated the position of the modern observed treeline based on July temperatures (=climate-analogue) for the years 2000-3000 CE. Therefore, the current July temperature at the treeline position is determined and tracked along the transect at each time step of the climate forcing, which is similar to the climatic velocity of Loarie et al. (2009).”

– Appendix 1 Figure 1:"Forest expansion dynamics along the four transects (columns) extrapolated based on July temperatures (climate–analogue) for climate data based on RCP scenarios (rows). What do the negative values represent in migration rates? And why are the rates averaged beginning with 2000 instead of averaged over the particular intervals? Intervals would seem more informative: 2000–2100, 2100–2300, 2000–2500. indicate unit of y–axis (or provide/explain in legend) and unit of migration rate. Treeline position in each panel – is this the northernmost treeline position identified and the corresponding year?

Negative values represent a receding treeline and stem from calculating the mean over yearly migration rates in the time periods and ending at a more southerly position than at the start.

We selected intervals beginning in 2000 until 2100, 2300 or 2500 for the purpose of comparing whether, over longer time periods, the first strong migration is offset and returns back to a more southerly position.

The y-axis unit is provided for all plots at the left margin “Treeline position [km]”.

The unit of the migration rate was added in the caption along with the statement that the treeline position is stated at the top in each panel “In each panel the maximum treeline position and the corresponding year is provided next to migration rates in three periods in km decade-1.”

– Appendix 1 Figure 2: "Forest expansion dynamics along the four transects (columns) forced with climate data based on RCP scenarios (rows)." is green not treeline and dark green forestline? These definitions are quite hidden and come very late on line 251 while they are very important!

As stated above, we have corrected the definitions in the Methods section and in the captions.

– Appendix 2 Figure 1. Needs some text labeling each figure by transect. "Numbers are the first year when the simulated treeline position is equal to or farther north than the modern climate–analogue position." I see no numbers other than the RCP scenario identifiers. No years indicated, no blue color, seems to be different graph. Should also be moved to Appendix 2 section.– Appendix 2 Figure 2. The caption references Figure 2 but that seems an error. If it is not, please tell the reader what exactly they should look for in Figure 2. no red in figure mentioned in legend.

The figures and its legend were unfortunately misplaced and the caption for Appendix 2 Figure 1 referred to Appendix 2 Figure 2. We moved the panel plot of all four regions to the main part of the manuscript. In this new graph we added the name of the regions to each header of the individual plots.

– Finally, some references to other individual based models like Rupp, T.S., Chapin, F.S. and Starfield, A.M., 2001. Modeling the influence of topographic barriers on treeline advance at the forest–tundra ecotone in northwestern Alaska. Climatic change, 48(2), pp.399–416.

We added the citation in the first paragraph in the Introduction as it is an early simulation study similar to our setup, however it is not an individual based model, but is spatially-explicit in the sense that for seed dispersal 2x2 km grid cells are considered, and on each grid cell, processes capture the turnover of ecotypes, which is quite different from our approach of simulating each single tree from seedling stage onwards and from purely spatially-explicit processes such as seed dispersal and competition etc.